# Dynamic stroma reorganization drives blood vessel dysmorphia during glioma growth

Thomas Mathivet[1,2,*] (ID), Claire Bouleti[1,2], Matthias Van Woensel[3], Fabio Stanchi[1,2], Tina Verschuere[3], Li-Kun Phng[1,2,4], Joost Dejaegher[3], Marly Balcer[1,2], Ken Matsumoto[1,2], Petya B Georgieva[1,2], Jochen Belmans[3], Raf Sciot[5], Christian Stockmann[6], Massimiliano Mazzone[7,8] (ID), Steven De Vleeschouwer[3,9] & Holger Gerhardt[1,2,10,11,**] (ID)

## Abstract

Glioma growth and progression are characterized by abundant development of blood vessels that are highly aberrant and poorly functional, with detrimental consequences for drug delivery efficacy. The mechanisms driving this vessel dysmorphia during tumor progression are poorly understood. Using longitudinal intravital imaging in a mouse glioma model, we identify that dynamic sprouting and functional morphogenesis of a highly branched vessel network characterize the initial tumor growth, dramatically changing to vessel expansion, leakage, and loss of branching complexity in the later stages. This vascular phenotype transition was accompanied by recruitment of predominantly pro-inflammatory M1-like macrophages in the early stages, followed by *in situ* repolarization to M2-like macrophages, which produced VEGF-A and relocate to perivascular areas. A similar enrichment and perivascular accumulation of M2 versus M1 macrophages correlated with vessel dilation and malignancy in human glioma samples of different WHO malignancy grade. Targeting macrophages using anti-CSF1 treatment restored normal blood vessel patterning and function. Combination treatment with chemotherapy showed survival benefit, suggesting that targeting macrophages as the key driver of blood vessel dysmorphia in glioma progression presents opportunities to improve efficacy of chemotherapeutic agents. We propose that vessel dysfunction is not simply a general feature of tumor vessel formation, but rather an emergent property resulting from a dynamic and functional reorganization of the tumor stroma and its angiogenic influences.

**Keywords** glioma; live imaging; myeloid cells; VEGF; vessel dysmorphia

**Subject Categories** Cancer; Neuroscience; Vascular Biology & Angiogenesis

See also: **M Lohela & K Alitalo** (December 2017)

## Introduction

Angiogenesis, the formation of new blood vessels from preexisting ones (Risau, 1997), establishes oxygen supply to growing tumors and is initiated by hypoxia-induced expression of pro-angiogenic factors, such as vascular endothelium growth factor (VEGF). In contrast to developmental and physiological angiogenesis, tumor angiogenesis generates a chaotic blood vessel network, featuring tortuous, dilated, and non-hierarchically organized vessels (Bergers & Benjamin, 2003; Baluk *et al*, 2005). Current concepts suggest that this dysmorphia of tumor vessels causes irregular blood flow, vascular leakage, edema formation, and limits systemic delivery of chemotherapeutics to the tumor (Carmeliet & Jain, 2011).

Tumor-associated macrophages are a major component of the tumor stroma (Qian & Pollard, 2010; Noy & Pollard, 2014), and their accumulation correlates with poor prognosis for patients (Zhang *et al*, 2012). Among multiple roles in tumor progression, macrophages have been shown to modulate angiogenesis (Zumsteg & Christofori, 2009; Qian & Pollard, 2010) and to limit the efficacy of anti-tumoral therapies (De Palma & Lewis, 2013). In gliomas, myeloid cell depletion using various approaches has led to

1  Vascular Patterning Lab, Center for Cancer Biology, VIB, Leuven, Belgium
2  Vascular Patterning Lab, Department of Oncology, KU Leuven, Leuven, Belgium
3  Department of Neurosciences, Laboratory of Experimental Neurosurgery and Neuroanatomy, KU Leuven, Leuven, Belgium
4  Laboratory for Vascular Morphogenesis, RIKEN Center for Developmental Biology, Kobe, Japan
5  Department of Pathology, KU Leuven and University Hospitals Leuven, Leuven, Belgium
6  UMR 970, Paris Cardiovascular Research Center, Institut National de la Santé et de la Recherche Médicale (INSERM), Paris, France
7  Lab of Molecular Oncology and Angiogenesis, Department of Oncology, KU Leuven, Leuven, Belgium
8  Lab of Molecular Oncology and Angiogenesis, Center for Cancer Biology, VIB, Leuven, Belgium
9  Department of Neurosciences, KU Leuven, Leuven, Belgium
10 Integrative Vascular Biology Laboratory, Max-Delbrück-Center for Molecular Medicine, Helmholtz Association (MDC), Berlin, Germany
11 Berlin Institute of Health (BIH), Berlin, Germany
  *Corresponding author. Tel: +33 153988018; E-mail: thomas.mathivet@inserm.fr
  **Corresponding author. Tel: +32 16373220; E-mail: holger.gerhardt@kuleuven.vib.be

contradictory results: On one hand, genetic or local pharmacogenetic ablation of macrophage populations slows down glioma progression, suggesting a tumor supportive role, potentially through modulation of angiogenesis (De Palma et al, 2005; Zhai et al, 2011). On the other hand, pharmacological depletion of macrophages using ganciclovir reportedly results in accelerated tumor growth with minor effects on vessel density (Galarneau et al, 2007).

Current limitations in the resolution of non-invasive in vivo imaging technologies do not allow a detailed follow-up of vascular patterning in glioma in a time-dependent manner. The poor accessibility of glioma tissue to intravital light microscopy in experimental models, and the chaotic nature of tumor vessels provide substantial challenges to our ability to resolve vascular patterning mechanisms in glioma angiogenesis at the cellular level. Therefore, the questions precisely when and where which macrophage populations influence tumor vessel patterning, and how, require further investigation.

Here, we used a surgical cranial window model (Ricard et al, 2014) and high-resolution two-photon microscopy, to elucidate the details of blood vessel morphogenesis during glioma progression in vivo. We demonstrate that contrary to current concepts, initial tumor blood vessel formation is highly functional, resulting in homogenous caliber, branching blood vessels with barrier properties and effective perfusion. During tumor growth, blood vessel patterning subsequently deteriorates with branching profoundly reduced at the expense of increased vessel diameter associated with recruitment and polarization of VEGF producing alternative activated M2-like macrophages that cluster tightly around blood vessels. We propose that vessel abnormalization limiting chemotherapeutic efficacy is driven by infiltrating and M2-polarizing macrophages, which redistribute VEGF bio-availability and alter its gradient in the tumor microenvironment.

# Results

### Progressive blood vessel dysmorphia during glioma growth

In order to visualize changes in the vasculature during glioma progression, we implemented a glioma mouse model adapted for intravital imaging by multiphoton microscopy (Ricard et al, 2014). In our adaptation of this model, spheroids of syngenic C57Bl6 mouse CT2A (Seyfried et al, 1992; Oh et al, 2014) and GL261 (Szatmari et al, 2006) glioma cells modified to stably express blue fluorescent protein TagBFP (Subach et al, 2008) were engrafted in the endothelial-specific Pdgfb-iCre line (Claxton et al, 2008) and in Cdh5-iCre (Sorensen et al, 2009) crossed with the ROSA$^{mT/mG}$ reporter mouse (Muzumdar et al, 2007). Tamoxifen-induced activation of membrane-targeted GFP (mG) in brain endothelial cells was performed 10 days before tumor implantation, allowing us to follow blood vessel growth over time. Other non-tumor and non-endothelial cells derived from the host expressed membrane-targeted red fluorescent protein tdTomato (mT), enabling to simultaneously detect also pericytes, macrophages, and glial cells.

Hypoxia is one of the main triggers of tumor angiogenesis (Maxwell et al, 1997; Stoeltzing et al, 2004; Jensen et al, 2006). Indeed, regions in the glioma tumor core upregulated hypoxia responsive genes including glut1 (Fig 1A) and hif1α (Fig EV1A). Timelapse imaging identified active dynamic tip cell sprouting, confirming the highly angiogenic nature of the tumor environment

(Fig 1B, Movie EV1). However, over time we observed a progressive deterioration of blood vessel patterning. Whereas early vessel formation (2-week tumor growth) exhibited hallmarks of sprouting angiogenesis, forming normal caliber vessels and regular branching similar to the healthy contralateral hemisphere, vessels at late stages (5 weeks) showed significantly reduced branching but profoundly increased vessel diameter (average threefold) (Figs 1C and D, and EV1C). This apparent loss of vessel size control and loss of branching complexity (Fig 1D and E) during progressive tumor growth (Fig EV1D) was associated with altered perfusion of the blood vessel network, identified by FITC-dextran perfusion (Fig 1F and G). In addition, Evan's blue extravasation illustrated blood vessel leakiness at late-stage tumor growth (Fig 1H and I). Endothelial specificity of Pdgfb-iCre recombination was confirmed by CD31 co-staining (Fig EV1B) and showed a very high percentage of endothelial cell recombination in this brain tumor model. Cre-expression from the second endothelial cell-specific line Cdh5-iCre also confirmed selectivity, but with lower recombination efficiency (Fig EV1C).

### Blood vessel dysmorphia coincides with macrophage recruitment

Studying the tumor microenvironment in search for factors that could explain blood vessel changes at late-stage glioma growth, we observed concomitant recruitment of host cells (mT positive) located in close proximity to blood vessels in late-stage tumor growth. Immunolabeling identified these cells as F4/80-positive macrophages (Fig 2A). More selective markers for innate or adaptive immunity cell types confirmed a major recruitment of macrophages, but very few T cells, B cells, natural killer cells, neutrophils, and dendritic cells (Appendix Fig S1A–F). To label and track myeloid cells, we crossed the Csf1r-Mer-Cre-Mer mouse line (Qian et al, 2011) with the ROSA$^{mT/mG}$ reporter mouse line (Muzumdar et al, 2007), which, following tamoxifen injection, allows the visualization of mG-positive myeloid cells. In vivo imaging during glioma progression at early (2 weeks)- and late-stage growth (4 weeks) confirmed the recruitment of Csf1r-Cre reporter (mG)- positive macrophages concomitant with the observed blood vessel diameter increase (Movies EV2 and EV3). Analysis of the contralateral hemisphere revealed no GFP reporter-positive cells, indicating that the inducible Csf1r-Mer-Cre-Mer promoter drives specific recombination in recruited myeloid cells but not in brain resident macrophages and microglia (Appendix Fig S2A).

### Glioma-invading macrophages are bone marrow derived

Transplantation of LifeAct-GFP bone marrow (Riedl et al, 2010) into irradiated ROSA$^{mT/mG}$ reporter mice followed by tumor implantation revealed exclusively LifeAct-GFP-positive macrophages surrounding blood vessels at late tumor growth, demonstrating bone marrow rather than brain-derived local macrophage sources (Fig 2B). The transplanted LifeAct-GFP-positive macrophages spread into the tumor in early glioma growth but were later located close to blood vessels with a threefold decrease of macrophage to blood vessel distance from early- to late-stage tumor growth (Fig 2C). Moreover, LifeAct-GFP-positive macrophages showed increased proliferation during late-stage tumor growth as identified by an approximately twofold increase in Ki67-positive macrophages compared to early stage (Appendix Fig S3). To confirm that LifeAct-GFP was only expressed in bone marrow-derived macrophages and not resident

**Figure 1. Blood vessel abnormalities arise during progressive glioma growth.**

A   Glucose transporter1 (Glut1) immunohistochemistry on sections of 5-week growth glioma in ROSA$^{mT/mG}$::*Pdgfb-iCre* mouse (50-µm depth stack). Hypoxic tumor cells upregulate Glut1.

B   Still image of two-photon live imaging on 2-week growth glioma implanted in ROSA$^{mT/mG}$::*Pdgfb-iCre* mouse demonstrating tip cell filopodia extension indicative of sprouting. See Movie EV1.

C   Representative images of two-photon live imaging of the same glioma area of the same mouse on 2- and 5-week growth glioma (BFP positive) implanted in ROSA$^{mT/mG}$::*Pdgfb-iCre* mouse (350-µm depth stack). Note differences in network complexity and vessel diameter.

D   Blood vessel diameter quantification: sprouting blood vessels at 2-week growth present caliber comparable to vessels in the healthy brain (hb). At 5-week growth, tortuous blood vessels in the tumor are significantly more dilated than in the healthy brain ($n$ = 8 mice).

E   Branchpoint quantification during glioma growth ($n$ = 8 mice).

F   Two-photon live imaging of vessel perfusion using FITC-dextran IV injection in ROSA$^{mT/mG}$ mice (asterisks correspond to unperfused vessel segment) (300-µm depth stack).

G   Quantification of unperfused vessel segments (asterisks in F) shows increase during glioma growth ($n$ = 5 mice).

H   Tumor-bearing brain overview images after Miles assay.

I   Late-stage glioma growth vessels are leaky compared to early tumor growth ($n$ = 5 mice per group).

Data information: Statistical analysis (D, E, G, I) one-way ANOVA followed by multiple comparisons Tukey's test. Error bars: mean ± SD. Scale bars: (A) 250 µm; (B) 50 µm; (C,F) 150 µm; (H) 0.5 cm.

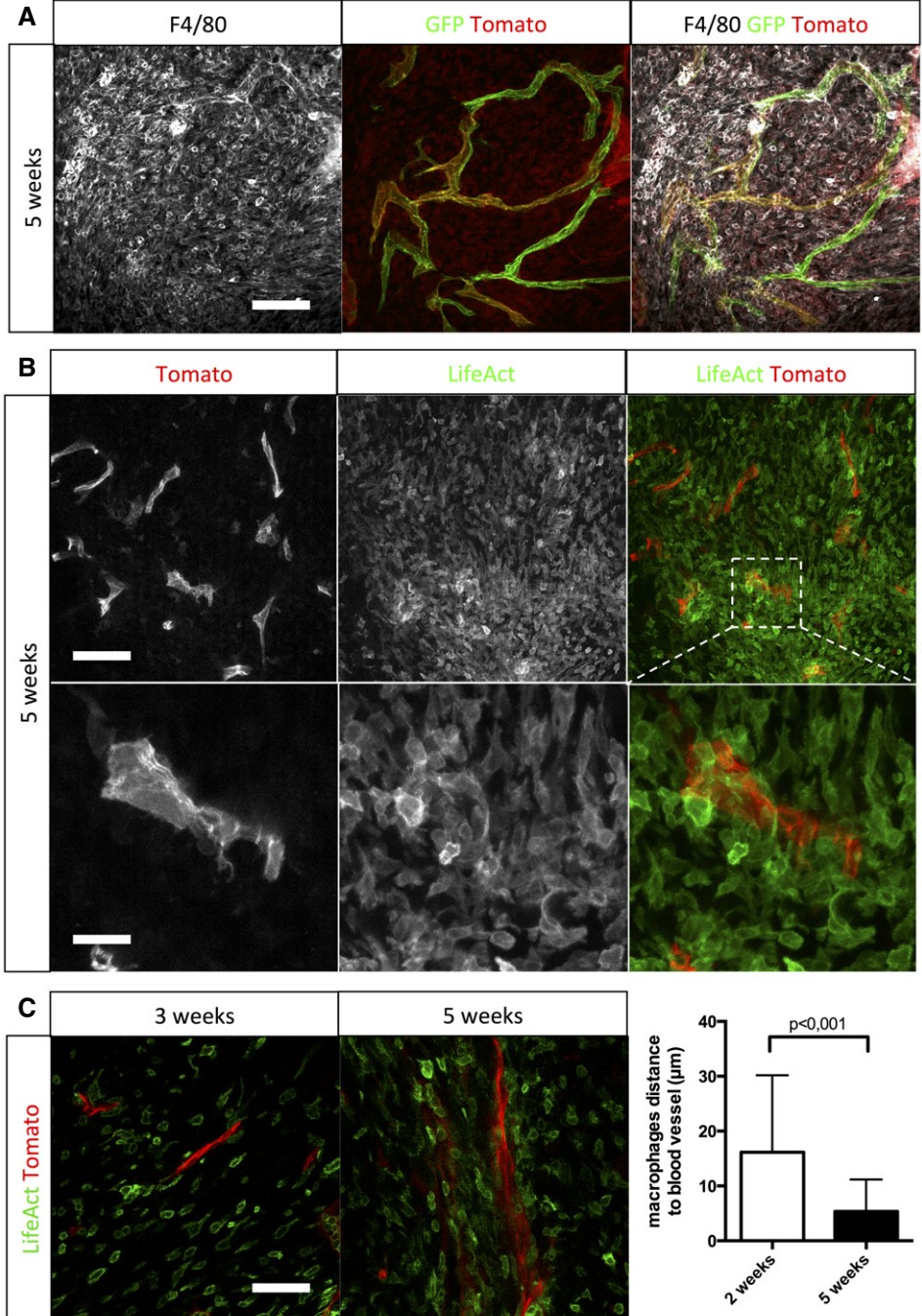

**Figure 2. Macrophages invading growing glioma are bone marrow derived.**

A  F4/80 immunohistochemistry on a section of 5-week growth glioma implanted in ROSA^mT/mG::*Pdgfb-iCre* mouse (50-μm depth stack).

B  Two-photon live imaging of LifeAct-GFP bone marrow transplantation in 5-week implanted glioma in ROSA^mT/mG mice (100-μm depth stack).

C  Two-photon live imaging of LifeAct-GFP bone marrow transplantation in 3- and 5-week implanted glioma in ROSA^mT/mG. Macrophages relocate close to blood vessels during glioma growth (50-μm depth stack; *n* = 4 mice).

Data information: Statistical analysis: (C) *t*-test. Error bars: mean ± SD. Scale bars: (A) 150 μm; (B) upper panels: 70 μm; (B) lower panels: 280 μm; (C) 100 μm.

macrophages, we imaged the contralateral hemisphere where no GFP-positive cells were detected (Appendix Fig S2B). To understand the contribution of resident brain macrophages/microglia in our glioma model, we performed Iba1 immunostaining (Appendix Fig S4A) and noticed that Iba1-positive cells were preferentially localized at the tumor margin and not, in contrast to the F4/80-positive bone marrow-derived macrophages (Appendix Fig S4B) in the tumor core (Appendix Fig S4C).

## Macrophages switch from M1 to M2 polarization during glioma growth

The role of macrophages in tumor biology depends on their activation state (Casazza & Mazzone, 2014). Circulating monocytes can either polarize as M1 cytotoxic macrophages promoting a Th1 immune response and as such have an anti-tumoral effect, or as M2 immunosuppressive macrophages that drive Th2 responses and thus have a pro-tumoral and pro-angiogenic effect through secretion of growth factors including VEGF (Schmid & Varner, 2010). The true heterogeneity of macrophages is likely even greater, with subcategories to the M1/M2 profile emerging (Auffray *et al*, 2009). Recent studies suggest that macrophages in the tumor microenvironment may be plastic, and possibly switch phenotype from M1 to M2 during tumor growth (Zaynagetdinov *et al*, 2011; Sica *et al*, 2012).

To analyze macrophage polarization in the glioma setting, we performed immunofluorescence labeling on thick sections at different stages of tumor growth. In early stages of tumor growth (2 weeks), 72% of the macrophages were MHCII$^{high}$ MRC1$^{neg}$ M1 macrophages and very few MHCII$^{low}$ MRC1$^{+}$ M2 macrophages were detected (Fig 3A and B). The opposite was observed in late-stage tumor growth (5 weeks) where 89% of the macrophages were M2 MHCII$^{low}$ MRC1$^{+}$ macrophages (Fig 3A and B). The specificity of immunolabeling using MHCII and MRC1 for M1 and M2 macrophages, respectively, was confirmed with double staining in which no overlap was observed between both populations (Appendix Fig S5). The polarization of macrophage populations was further assessed by flow cytometry analysis using Movahedi *et al* settings (Movahedi *et al*, 2010), confirming the overall switch of monocyte-derived cells to a M2 phenotype (Figs 3C and EV2). Interestingly, the number of MRC1-positive macrophages significantly correlates with blood vessel diameter increase during glioma growth, suggesting a direct effect of the switch in macrophage polarization on blood vessel morphology (Fig 3D).

These data are compatible with either an *in situ* switch within the tumor microenvironment from M1 to M2 macrophages or with a *de novo* recruitment of distinct populations. To distinguish between these possibilities, we injected fluorescently labeled anti-MHCII antibody systemically in 3-week glioma-bearing mice and sacrificed these animals at 6 h or 24 h post-injection. At the 6-h time-point, immunostaining on brain tumor sections revealed that macrophages in the tumor labeled with MHCII antibody were negative for MRC1 (Fig 3C). In contrast, at the 24-h time-point, 75% of the MHCII labeled macrophages in the tumor were also positive for MRC1 immunolabeling, indicating that in this 24-h time window, originally M1 macrophages positive for MHCII switched *in situ* to a M2 phenotype, expressing the MRC1 marker (Fig 3E).

## M2 macrophage polarity correlates with blood vessel dysmorphia in human glioma

To investigate the relevance of our findings in human pathology, we collected fresh glioma samples from 29 patients of WHO grade II to grade IV (Table EV1). Human glioma progression is characterized clinically by blood vessel alteration and leakiness (Appendix Fig S6). Performing immunostaining for Glut1 as a brain endothelial cell marker, we observed, as in the mouse model, a stage-related change

of blood vessel patterning with loss of network complexity and vessel diameter increase in high-grade gliomas (Fig 4A). Moreover, immunolabeling of total and M1/M2 macrophages revealed an intriguing similarity in phenotype and localization of macrophages between the stages of the mouse model and human glioma (Fig 4B–D). The number of M2 MRC1-positive macrophages significantly correlated with vessel diameter in accordance with the WHO glioma grade classification. These results would be compatible with the idea of a direct relationship between M2 macrophages accumulation and vessel mispatterning (Fig 4E).

## Macrophage depletion restores blood vessel caliber and function

In order to understand the potential of anti-macrophage therapy concerning blood vessel function and glioma progression, we inhibited macrophage recruitment in the tumor microenvironment using anti-CSF1 monoclonal antibody treatment. Systemic administration of anti-CSF1 antibody in mice has previously been shown to effectively inhibit macrophage differentiation, proliferation and survival (Evans *et al*, 1989; Hume & MacDonald, 2012). At late-stage growth (4 weeks), Glut1 and CD31 immunostaining revealed a profound "normalization" of blood vessel parameters in anti-CSF1 antibody-treated glioma-bearing mice compared to isotype-matched control antibody treatment (Fig 5A and B). A 50% decrease in the total population of macrophages (Fig 5C) was accompanied by a significantly reduced blood vessel caliber (Fig 5A), partially restored Glut1 endothelial expression, indicating improved barrier-function (Fig 5B), and restored vessel perfusion (Fig 5E). Unexpectedly, macrophage depletion and vessel normalization were associated with a significant twofold increase in tumor growth (Fig 5D). This could be either related to improved vessel function or directly to modulation/compensation of the immune response. To investigate a potential immune response compensation because of the absence of macrophages, we assessed neutrophil and myeloid-derived suppressor cells (MDSCs) populations in response to the treatment by immune-labeling for Ly6$^{G}$ or CD11B/Ly6$^{C/G}$, respectively. We observed a tendency of neutrophil accumulation (Appendix Fig S7A–C), but not MDSCs following anti-CSF1 treatment (Swierczak *et al*, 2014; Appendix Fig S7B–D). To further investigate the effects of macrophage accumulation on glioma vessel parameters, we treated early-stage tumor-bearing mice (3 weeks) with recombinant-CSF1 protein. Enhanced tumor macrophage recruitment (plus 53%) in early-stage tumors (Fig 6A and B) accelerated vessel dysmorphia and diameter increase (63%) compared to control carrier treatment (Fig 6A and C). In addition, recombinant CSF1 treatment significantly delayed glioma growth (52% size reduction) compared to control and anti-CSF1 antibody treatment (Fig 6D).

## M2 macrophage-derived VEGF drives vessel dysmorphia in late-stage glioma

We next asked whether and how the phenotypic macrophage switch could explain the vascular changes and dysmorphia. Given the significant correlation between M2 MRC1-positive macrophages and blood vessel dilation during glioma growth (Fig 3D), and the pronounced M2 accumulation around blood vessels, we searched for an M2-derived factor that might mediate the effects. We performed qPCR analysis of tumor macrophages after specific

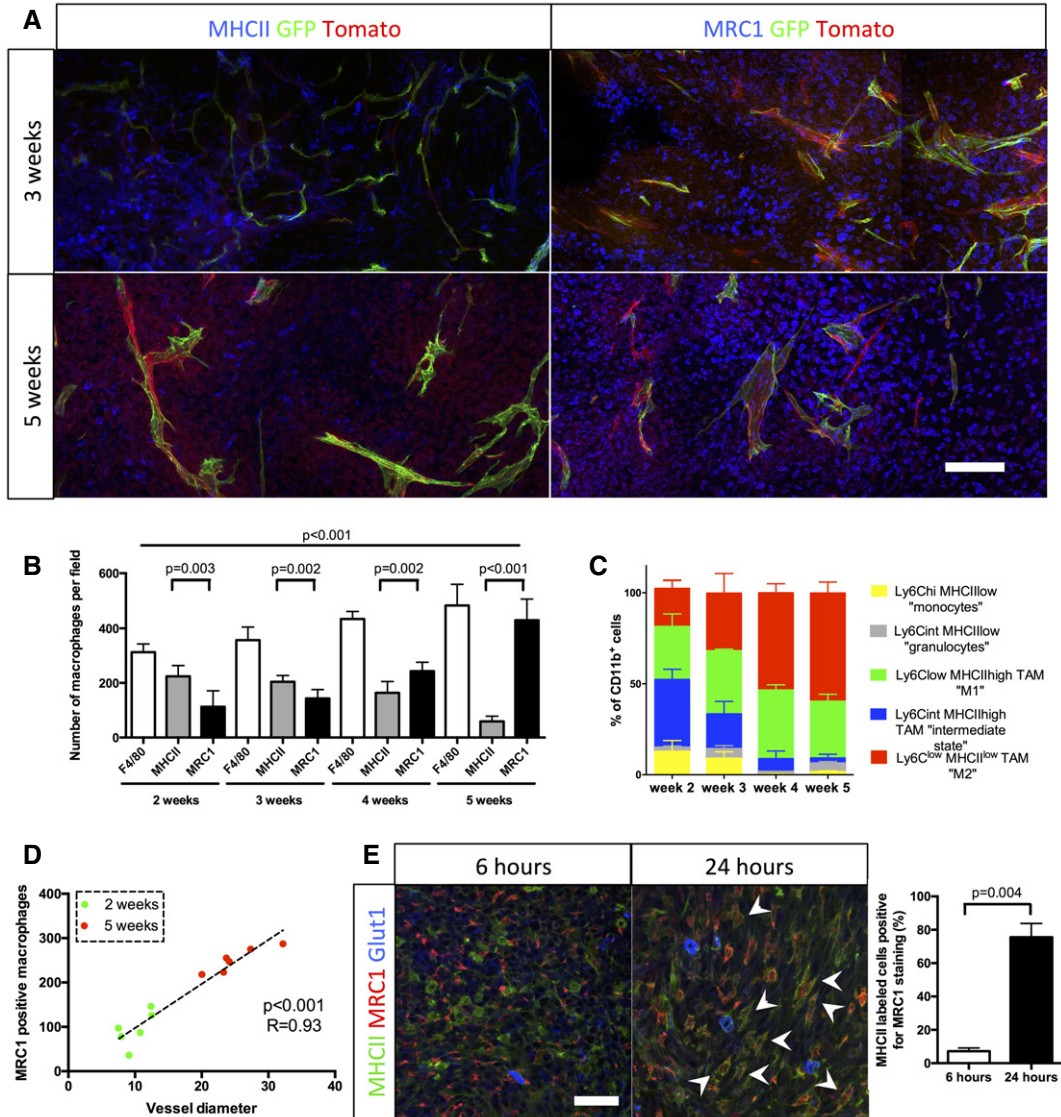

**Figure 3.  Concomitant macrophage accumulation and *in situ* switch from M1 to M2 phenotype with blood vessel abnormalities during glioma growth.**

A   MHCII and MRC1 immunohistochemistry on a section of 3- to 5-week growth glioma implanted in ROSA^mT/mG^::*Pdgfb-iCre* mouse. M1 macrophages, preferentially located in hypoxic area, are abundant at 3 weeks and their number decreases at 5 weeks. M2 macrophages, preferentially located around blood vessels, are present at a low level at 3 weeks and their number increases at 5 weeks.

B   Macrophage quantification during glioma growth (n = 5 mice per group).

C   MHCII and MRC1 macrophage population FACs analysis in 2-, 3-, 4-, and 5-week glioma (subsets in gated CD11b⁺ cells; n = 3 mice per group).

D   Correlation analysis of M2-like macrophages accumulation (MRC1 positive) with vessel dysmorphia over glioma growth (n = 12 mice).

E   MRC1 immunohistochemistry on 3-week growth glioma in ROSA^mTmG^::*Pdgfb-iCre* mouse following MHCII-FITC labeled IV injection at 6 or 24 h (5-μm depth stack). Note the segregation of MRC1 and MHCII cell labeling at 6 h and the overlapping (examples pointed with arrow heads) of these two markers at 24 h (n = 5 mice per group).

Data information: Statistical analysis: (B) one-way ANOVA followed by multiple comparisons Tukey's test; (D) Spearman's correlation test; (E) *t*-test. Error bars: mean ± SD. Scale bars: (A) 200 μm; (E) 50 μm.

isolation by magnetic-activated cell sorting of M1 or M2 macrophages using anti-MHCII or anti-MRC1-coated beads, respectively (Fig 7A). MHCII-sorted cells expressed more CCR7, CCL19, CXCL10, and IL12 than MRC1-sorted cells (Fig 7A). On the contrary, MRC1-sorted cells expressed significantly more TGFβ and VEGF-A, together with the M2 markers mannose receptor and CD209 (Fig 7A) (Bulnes *et al*, 2012). Given the cytokine profile of M2

macrophages and their perivascular accumulation in late-stage tumors, it is conceivable that presentation, localization, and bio-availability of VEGF-A will change during the transition from early- to late-stage tumors. Indeed, *in situ* hybridization (Fig EV3A) and receptor-body staining using soluble VEGFR1 (sFlt1) as a probe (Figs 7B and EV3B) identified production and presentation of a larger amount of VEGF-A in the direct vicinity of endothelial cells

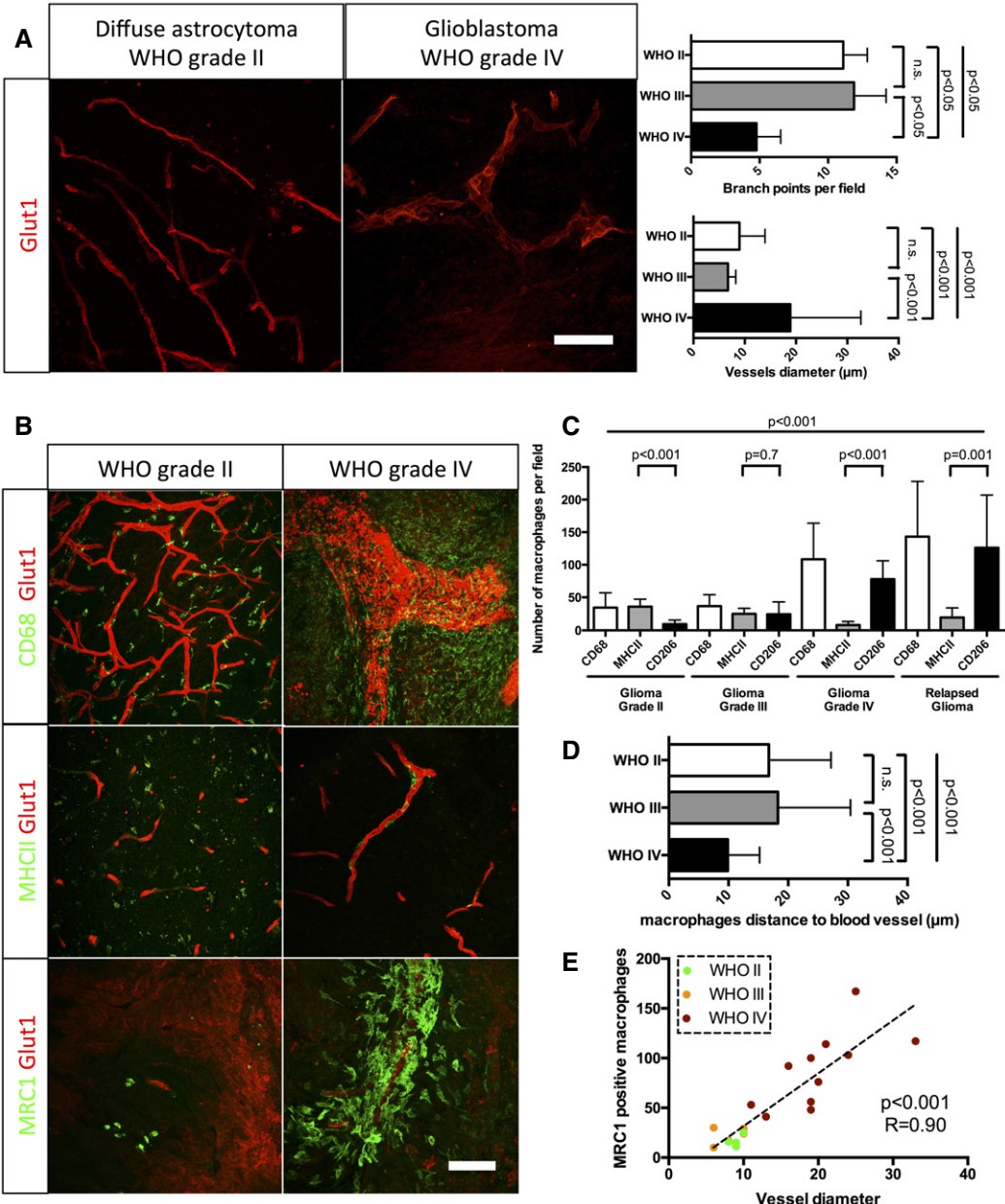

**Figure 4. Blood vessel dysmorphia is associated with macrophage polarization in progressive grades of human glioma.**

A   Glut1 immunohistochemistry on human grade II and grade IV glioma samples (50-μm depth stack). Blood vessel diameter increases and network complexity decreases with increasing grade.

B   CD68, MHCII, and MRC1 immunohistochemistry on human grade II and grade IV glioma samples (50-μm depth stack).

C   Macrophage polarization quantification. Low-grade glioma is characterized by high number of M1 polarized macrophages. High-grade glioma is characterized by high number of M2 polarized macrophages.

D   Correlation analysis of M2-like macrophages accumulation (MRC1 positive) with vessel dysmorphia depending on the tumor grade.

E   Quantification of macrophage distance to the nearest blood vessel. Macrophages relocate close to blood vessels in late-stage gliomas. For number of patients per group, refer to Table EV1.

Data information: Statistical analysis: (A, C, D) one-way ANOVA followed by multiple comparisons Tukey's test; (E) Spearman's correlation test. Error bars: mean ± SD. *n* = 5 WHO grade II patients, *n* = 5 WHO grade III patients and *n* = 19 grade IV patients. Please refer to Table EV1. Scale bars: (A, B) 100 μm.

(Fig EV3C). Too much VEGF and a lack of gradient have been shown to stall sprouting while driving proliferation (Ruhrberg *et al*, 2002; Gerhardt *et al*, 2003). In addition, our recent work on the function of the VEGF-Dll4/Notch-VEGFR2 feedback loop in sprouting angiogenesis predicted that endothelial cells over-stimulated with VEGF and lacking a VEGF-gradient become synchronized (Bentley *et al*, 2008, 2009, 2014; Ubezio *et al*, 2016) and try to sprout at the same time, extending filopodia in every

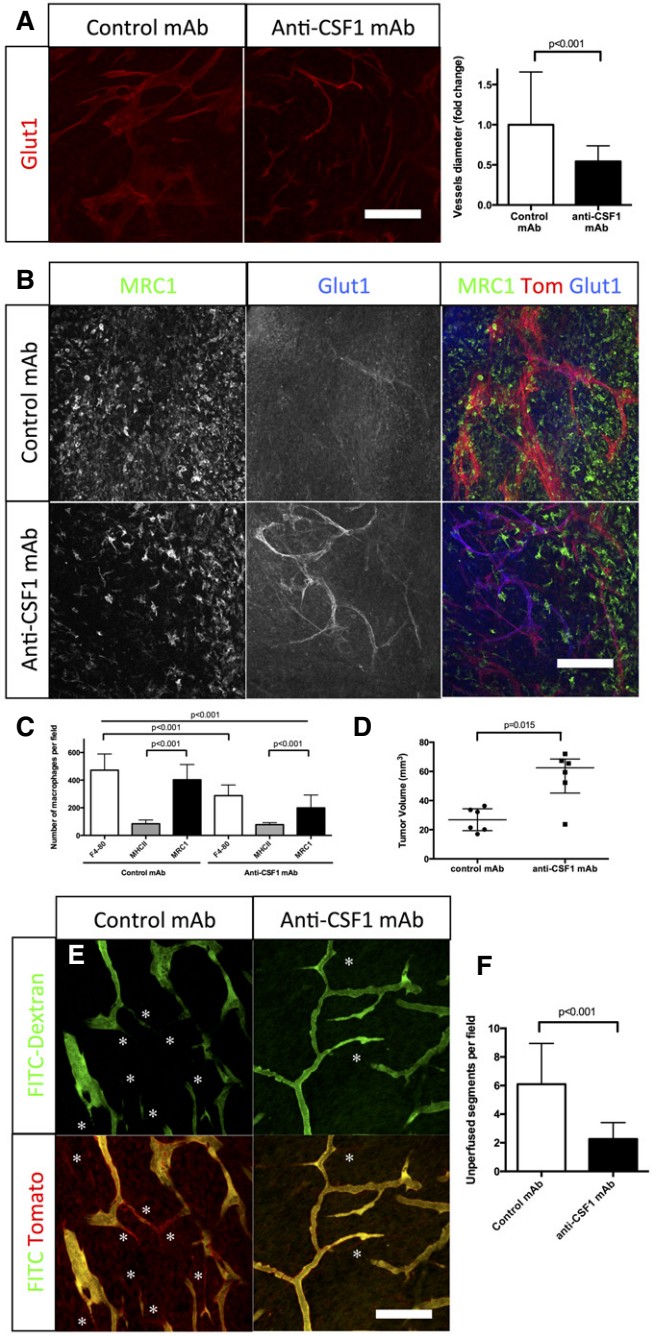

Figure 5.   Inhibition of macrophage recruitment restores patterning and functionality of the blood vessel network in glioma.

A   Glut1 immunohistochemistry on sections of 4-week growth glioma in ROSA[mT/mG] mice treated with anti-CSF1 or control antibodies. Anti-CSF1 antibody treatment induces a twofold decrease in vessel caliber compared to control antibody ($n = 6$ mice per group) (50-μm depth stack).
B   Glut1 and MRC1 immunohistochemistry on 4-week growth glioma in ROSA[mT/mG] mice treated with anti-CSF1 or control antibodies. Anti-CSF1 antibody induces a decrease in macrophage recruitment in the tumor microenvironment associated with an improved blood brain barrier functionality assessed by a better Glut1 blood vessel coverage compared to control antibody-treated mice (50-μm depth stack).
C   Macrophage quantification in response to anti-CSF1 or control antibody treatment ($n = 6$ mice per group).
D   Tumor volume quantification in response to anti-CSF1 or control antibodies ($n = 6$ mice per group).
E   Two-photon live imaging of vessel perfusion in 4-week implanted glioma growth using FITC-dextran IV injection in ROSA[mT/mG] mice treated with anti-CSF1 or control antibodies (asterisks correspond to unperfused vessel segments).
F   Quantification of unperfused vessel segments. Anti-CSF1 antibody treatment induces a reduction in the number of unperfused vessel segments (asterisks in E). ($n = 5$ mice per group) (300-μm depth stack).

Data information: Statistical analysis: (A, F) $t$-test; (C) one-way ANOVA followed by multiple comparisons Tukey's test; (D) Mann–Whitney $U$-test. Error bars: (A, C, F) mean ± SD; (D) median interquartile. Scale bars: (A, B, E) 150 μm.

## Macrophage-derived VEGF depletion restores blood vessel caliber and function

To study the contribution of M2 macrophages derived VEGF in vascular dysmorphia, we analyzed the *Vegfa[fl/fl]::LysMCre* mice (Fig 7C–G) (Stockmann *et al*, 2008). Depletion of VEGF expression in 61% of the myeloid cells invading the tumor revealed by soluble Flt1 binding (Fig 7E) did not alter the total number of F4/80-positive macrophages (Fig 7D). However, blood vessel caliber was decreased by 53%, demonstrating the involvement of macrophage-derived VEGF in blood vessel alteration (Fig 7A and F). Remarkably, the myeloid cell VEGF depletion was accompanied by a significant 25% increase in tumor growth (Fig 7G), suggesting that the macrophage-induced vessel dysmorphia driven by VEGF limits rapid tumor expansion.

Moreover, administration of a soluble VEGF trap using sFlt1 also normalized the glioma vasculature (Fig EV4A) and altered macrophage recruitment in the tumor microenvironment, potentially by interfering with the VEGFR1/VEGF pathway in macrophages (Fig EV4B and C). Interestingly, however, this did not affect tumor growth, showing that complete VEGF downregulation in the tumor does not have the same effect as the specific depletion of VEGF from perivascular M2 macrophages (Fig EV4D).

In order to analyze the specificity of vessel dilation for glioma and its link to myeloid cell recruitment in the tumor microenvironment, we inoculated B16 melanoma in the flank of wild-type mice and depleted macrophages using anti-CSF1 mAb treatment (Appendix Fig S9). As observed during glioma growth, melanoma irrigating blood vessels significantly expanded over time during tumor growth (Appendix Fig S9A and B). This vessel dilation correlated with an accumulation of M2 macrophages in late-stage tumor growth (Appendix Fig S9D). Depletion of macrophages nevertheless induced a reduction of vessel caliber (Appendix Fig S9A–C), yet with only minor effects on tumor volume (Appendix Fig S9C).

direction leading to vessel diameter expansion. Live imaging showed iterative bursts of lateral filopodia emanating from the enlarged vessels, but failing to produce productive sprouts (Movie EV4). Moreover, VEGF-A is also a potent vascular permeability factor. The presentation of VEGF-A by M2 macrophages in close contact with endothelial cells will thus likely cause or contribute to the vascular leakage and perfusion defects observed in late-stage gliomas (Fig 1H and I). As a consequence, sequestration of blood vessels by macrophages is a likely mechanistic cause for the observed hypoxic area expansion during glioma progression as highlighted by Glut1 immunostaining (Appendix Fig S8).

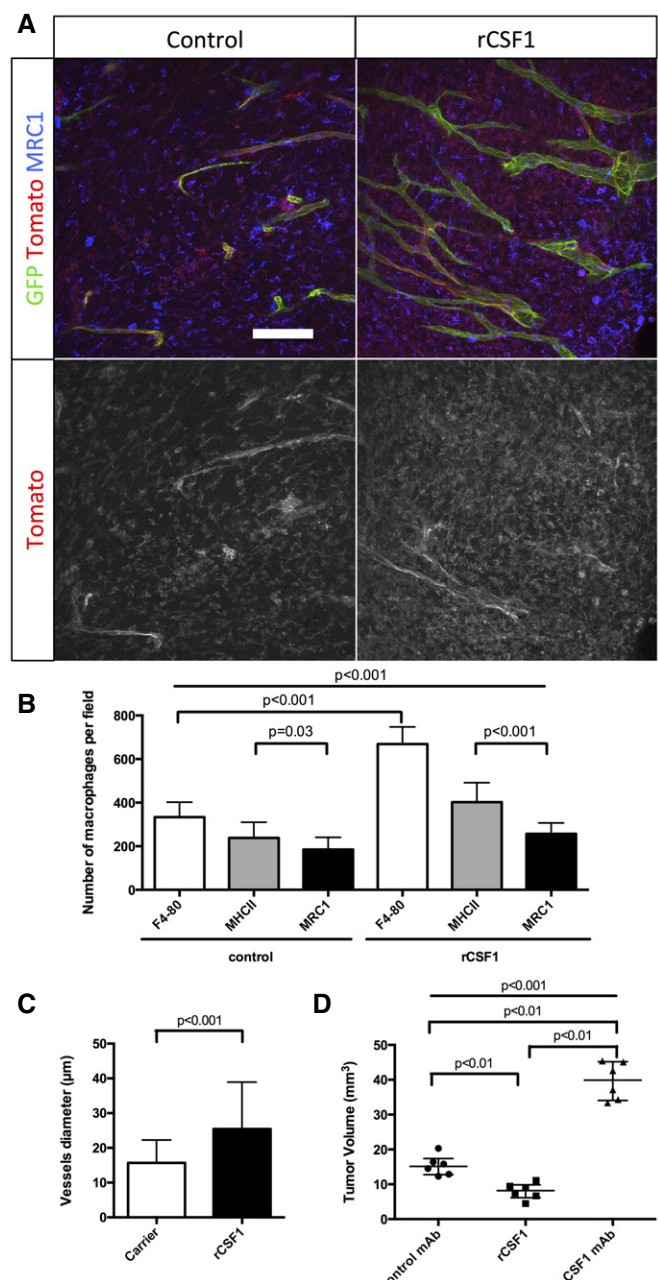

**Figure 6.  Promoting macrophage recruitment in glioma drives early dysmorphia of the tumor blood vessel network.**

A   MRC1 immunohistochemistry on sections of 3-week growth glioma in ROSA$^{mT/mG}$ mice treated with recombinant CSF1 or carrier. Recombinant CSF1 treatment induces blood vessel enlargement in early tumors (50-μm depth stack).

B   Quantification of Tomato-positive macrophages. Recombinant CSF1 treatment induces increased macrophage recruitment compared to control (*n* = 6 mice per group).

C   Vessel diameter quantification. Recombinant CSF1 treatment drives blood vessel diameter increase compared to control (*n* = 6 mice per group).

D   Tumor volume quantification in 3-week glioma implanted mice treated with recombinant CSF1, anti-CSF1 antibody, or control antibody. Recombinant CSF1 treatment reduces tumor growth compared to control (*n* = 6 mice per group).

Data information: Statistical analysis: (B, D) one-way ANOVA followed by multiple comparisons Tukey's test; (C) *t*-test. Error bars: (B, C) mean ± SD; (D) median interquartile. Scale bar: (A) 100 μm.

models, temozolomide treatment significantly improved median survival compared to control mice (Fig 8B; Plowman *et al*, 1994; Friedman *et al*, 1995) without side effects on the immune cell compartment in the bone marrow (Fig EV5E). Interestingly, anti-CSF1 monoclonal antibody treatment combined with temozolomide treatment significantly potentiated the chemotherapeutic agent efficacy with a further increase in the median survival compared to mice treated with chemotherapeutic agent alone (Fig 8B). In order to understand the cause of combination therapy survival benefit, we performed immunostaining against activated caspase3 (Fig 8C–E) and phospho-H2AX (Fig EV5A and B) used as a tumor DNA damage marker. We observed a significant threefold increase in tumor cell death in the anti-CSF1 and chemotherapy combined treatment compared to chemotherapeutic treatment alone. Moreover, glioma cell death was homogenous in the entire tumor when mice were treated with the combination therapy compared to a patchy glioma cell death pattern in the chemotherapy treatment alone suggesting that improved vessel patterning (Fig 8D) and function allows a better overall tumor targeting. In line with this idea of vascular network functionality improvement, tumor hypoxia highlighted by Glut1-positive tumor cells area was reduced in response to TMZ and this effect was exacerbated following TMZ plus anti-CSF1 co-treatment (Fig EV5C and D).

## Discussion

Vascular endothelium growth factor derived from tumor-associated macrophages has been shown to drive the formation of a high-density vessel network in other mouse solid tumor models such as breast cancer (Stockmann *et al*, 2008). Also, bone marrow-derived cells (BMDCs) have been reported to invade glioma in response to hypoxia and stimulate neovascularization (De Palma *et al*, 2007; Du *et al*, 2008; Kioi *et al*, 2010). However, these fixed time-point and mostly late-stage studies implied that tumor vessels already form as tortuous and leaky conduits.

Our observations, using an *in vivo* longitudinal imaging approach, instead argue that vascular network dysmorphia is progressive and connected to changing stromal cell populations arising during late-stage growth. Interestingly, the combined vessel

### Macrophage depletion enhances chemotherapeutic agent efficacy

To investigate the relevance of our findings in a therapeutic treatment-based approach, we performed survival experiments combining temozolomide (TMZ) recommended chemotherapeutic agent for glioma treatment (Furnari *et al*, 2007; Sathornsumetee *et al*, 2007) and anti-CSF1 monoclonal antibody treatment (Fig 8A). For these endpoint survival studies, we adopted the classical cell suspension injection model of glioma we previously used (Van Woensel *et al*, 2017). Anti-CSF1 monoclonal antibody treatment alone did not significantly affect animal survival compared to the control group (Fig 8B). As previously shown in these glioma mouse

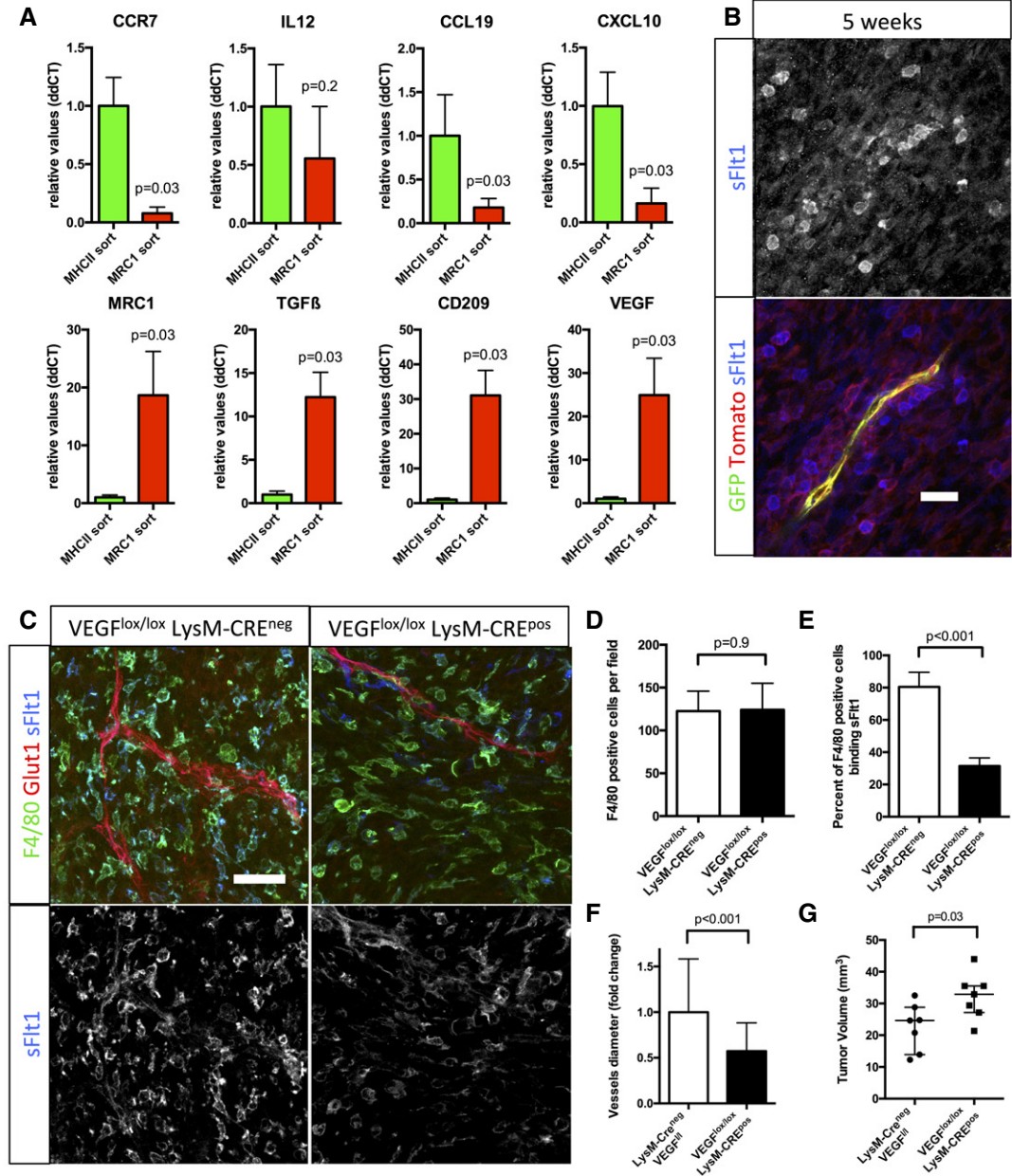

**Figure 7.  Macrophage-derived VEGF depletion restores blood vessel architecture in glioma.**

A  qPCR analysis of CCR7, IL12, CCL19, CXCL10, MRC1, TGFβ, CD209, and VEGF on RNA samples from MHCII or MRC1 tumor extracted macrophages (*n* = 5 isolation per group).

B  sFlt1 binding assay on 5-week growth glioma section implanted in ROSA^mT/mG::Pdgfb-iCre mouse. At late-stage tumor growth, M2 macrophages surrounding blood vessels present very high amount of VEGF (binding sFlt1) to the neighboring endothelial cells (5-μm depth stack).

C  F4/80 immunohistochemistry and sFlt1 binding on sections of 4-week growth glioma in *Vegfa*^fl/fl::*LysMCre*^+ or *Vegfa*^fl/fl ::*LysMCre*^− mice (50-μm depth stack).

D  Quantification of F4/80-positive macrophages. VEGF depletionin myeloid cells does not alter macrophages recruitment (*n* = 7 mice per group).

E  Quantification of F4/80-positive macrophages binding sFlt1. VEGF depletion is effective in 61% of the macrophages invading the tumor (*n* = 7 mice per group).

F  Vessel diameter quantification. Myeloid cell-derived VEGF depletion induces blood vessel "normalization" compared to control (*n* = 7 mice per group).

G  Tumor volume quantification in 4-week growth glioma in *Vegfa*^fl/fl::*LysMCre*^+ or *Vegfa*^fl/fl::*LysMCre*^− mice (*n* = 7 mice per group).

Data information: Statistical analysis: (D–F) *t*-test; (A, G) Mann–Whitney *U*-test. Error bars: (A, D–F) mean ± SD; (G) median interquartile. Scale bars: (A, C) 100 μm.

patterning and tumor growth data suggest that vessel dysmorphia can be disconnected from the actual size or growth of the tumor cell population. Moreover, the distinct transition in vessel patterning occurring between early- and late-stage glioma growth in the mouse model, similarly characterized human samples of different WHO malignancy grades. Our sequential *in situ* labeling of MHCII followed by post-fixation MRC1 labeling indicates that this progressive stroma alteration comprises a repolarization of M1-like to M2-like macrophages within the tumor. Moreover, this *in situ* repolarization to M2-like macrophages is accompanied by a

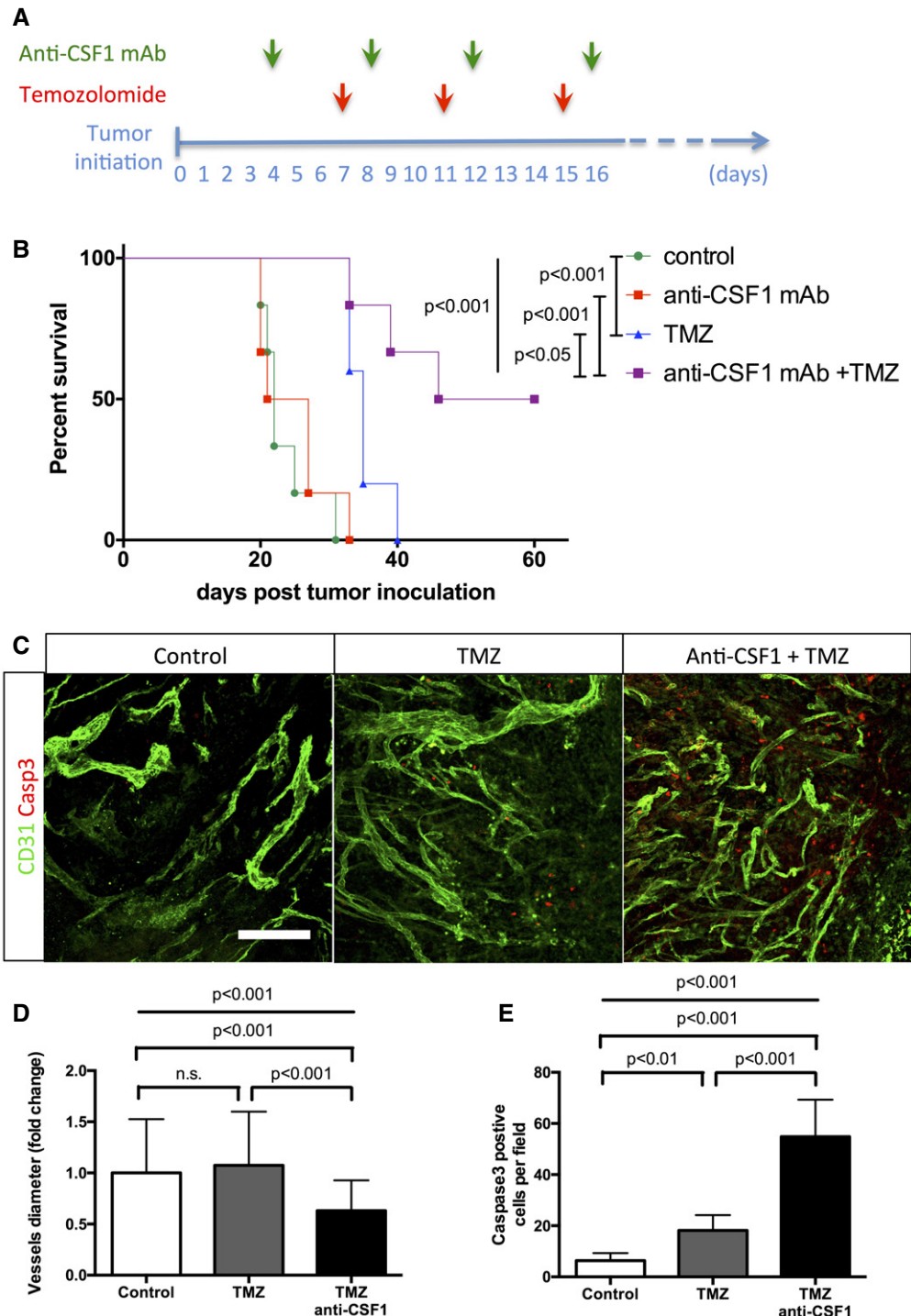

**Figure 8. Combining macrophage depletion with chemotherapy increases survival of glioma-bearing mice.**

A    Experimental design for the survival trial: 8-week-old wild-type mice were injected with GL261 cells to induce glioma formation and were assigned randomly to vehicle (*n* = 6), anti-CSF1 monoclonal antibody treatment (0.5 mg; *n* = 6), temozolomide treatment (40 mg/kg; *n* = 6), or temozolomide and anti-CSF1 mAb combination treatment (*n* = 6). Mice were weighed once daily until they developed symptoms or reached the trial endpoint.

B    Survival curves of the mice described in (A). Combination therapy is significantly more efficient than chemotherapy alone. (*n* = 6 mice per group).

C    phospho-H2AX and Glut1 immunohistochemistry on 3-week growth glioma in wild-type mice treated as described in (A).

D, E    Anti-CSF1 antibody in combination with temozolomide chemotherapeutic agent induces a wider tumor cell death efficiency with a significant decrease of blood vessel caliber (D) and an increase in caspase3-positive tumor cells (E) (50-μm depth stack) (*n* = 6 mice per group).

Data information: Statistical analysis: (B) Multiple comparisons Mantel–Cox log-rank; (D, E) one-way ANOVA followed by multiple comparisons Tukey's test. Error bars: mean ± SD. Scale bar: (C) 300 μm.

profound spatial rearrangement of these cells, relocating close to vessels to form perivascular clusters. Immunolabeling on sections and qPCR on isolated macrophage populations illustrated strong VEGF-A expression by M2 macrophages. Mechanistically, the relocation of M2 macrophages and their VEGF production to the immediate proximity of blood vessels could be the cause for reduced branching at the expense of increasing vessel diameter in late-stage tumors. Our previous studies identified elevated VEGF and disrupted gradients to synchronize neighboring endothelial cells through the VEGF-Dll4/Notch-VEGFR2 feedback loop (Ubezio et al, 2016). Such synchronization would be predicted to interfere with branching vessel morphogenesis by disrupting differential endothelial adhesion downstream of Notch (Bentley et al, 2014; Dejana & Lampugnani, 2014). Administration of a monoclonal antibody macrophage inhibitor targeting CSF1 maintains a functional and highly branched vasculature until the late-stage glioma. This suggests that the phenotypic switch of macrophages to the alternative activated M2 state contributes to the development of blood vessel dysmorphia during glioma growth. However, the mechanisms and outcomes of strategies targeting CSF1 or CSF1R are a matter of ongoing debate. Indeed, a number of recent studies report that macrophages stimulate tumor angiogenesis in a large variety of models and their depletion is accompanied with reduced angiogenesis, attenuated tumor growth, and an improved response to chemo and radiotherapies (DeNardo et al, 2011; Mitchem et al, 2013; Xu et al, 2013; Hughes et al, 2015; Shiao et al, 2015). In gliomas, myeloid cell depletion (De Palma et al, 2005; Zhai et al, 2011) or CSF-1R inhibition (Pyonteck et al, 2013) has been shown to exert anti-tumor effects or accelerated tumor growth (Galarneau et al, 2007; Stockmann et al, 2008).

In most studies, anti-CSF1R antibody-based therapies were shown to decrease total macrophages population in various tumor models in mice and patients (Manthey et al, 2009; Patel & Player, 2009; Ries et al, 2014; Cotechini et al, 2015). Other reports however suggest that anti-CSF1R inhibitor appeared to re-educate macrophages rather than decreasing their numbers in mouse glioma, correlating with tumor size reduction (Pyonteck et al, 2013). In our experiments, targeting the CSF1R ligand CSF1 reduces the total number of bone marrow-derived macrophages (Evans et al, 1989; Hume & MacDonald, 2012). Importantly, effects from targeting CSF1, unlike those achieved when targeting the receptor, should not be confounded by potential activities of the second receptor ligand IL-34 (De et al, 2016). Therefore, the use of CSF1R and CSF1 inhibitors in glioma might have differential effects, as the downstream signaling events triggered by CSF1 and IL-34 are distinct (Chihara et al, 2010; Barve et al, 2013).

In our hands, targeting macrophage recruitment in the glioma microenvironment with anti-CSF1 treatment or decreasing macrophage VEGF production similarly accelerate tumor growth, but restore a functional and perfused blood vessel network that can support effective delivery of anti-tumoral drugs. Indeed, macrophage depletion treatment in combination with recommended chemotherapy significantly improved survival of glioma-bearing mice and significantly increased glioma cell death homogeneously within the tumor. Together with the improved structure and perfusion of vessels after macrophage depletion, these data suggest that the survival benefit of combination with chemotherapy is due to an improved delivery of temozolomide. However, in the absence of direct measurements of drug within the tissue, we cannot exclude alternative effects including altered pharmacodynamics in combination treatment. Importantly, although the pathogenesis of the tumor and its mutational landscape will arguably differ in the transplantable mouse tumor model compared to the in situ tumorigenesis and progression in human tumors, the actual phenotypic progression of the vascular network and the immune cell recruitment appear highly similar, providing important new insights into the longitudinal development of tumor vessel density and functionality.

# Materials and Methods

### Animal procedure and glioma implantation

C57bl6 ROSA$^{mTmG}$::Pdgfb-iCre mice or C57bl6 ROSA$^{mT/mG}$::Cdh5-iCre (8–12 weeks) were intraperitoneally injected once or three times, respectively, with 80 µg/g of tamoxifen 10 days prior to surgery. Craniotomy was realized by drilling a 5-mm circle in between lambdoid, sagittal, and coronal sutures of the skull on ketamine/xylazine anesthetized mice. A 250-µm diameter CT-2A or GL261 glioblastoma cells spheroid was injected in the cortex and sealed with a glass coverslip cemented on top of the mouse skull.

Early-stage glioma growth was defined as ≤3 weeks.

Late-stage glioma growth was defined as ≥4 weeks.

For anti-CSF1 5A1 monoclonal antibody (BioXCell) treatment, 1-week growth glioma-bearing mice were injected IP with 0.5 mg/week.

For rCSF1 (R&D Systems) protein treatment, 1-week growth glioma-bearing mice were injected IP with 2 µg of protein every other day.

For mouse recombinant sFlt1 (R&D Systems) protein treatment, 1-week growth glioma-bearing mice were injected IP with 2 µg of protein every other day.

Humane endpoint of the experiment was reached when the animal lost 15–20% of its original weight. The anesthetized mouse was then transcardially perfused with 2% PFA solution. The mouse brain was harvested and fixed overnight in 4% PFA at 4°C. For in situ hybridization, the brain was post-fixed in −20°C cold methanol. For immunocytochemistry, the brain was washed with PBS and sectioned at the vibratome (200-µm thickness sections).

Tumor volume was measured on serial 200-µm tumor sections of the whole tumor (obtained at the vibratome) under stereo-microscope using Leica software according to Cavalieri's principle.

The same implantation method was applied to C57Bl6 ROSA$^{mTmG}$::Csf1r-Mer-iCre-Mer mice with intraperitoneal injection of 80 µg/g of tamoxifen weekly over the course of the tumor progression.

The same implantation method was applied to Vegf$^{lox/lox}$ LysMCre mice. For our study, seven Vegfa$^{fl/fl}$::LysMCre$^+$ female mice (8–12 weeks) were compared to seven Vegfa$^{fl/fl}$::LysMCre$^-$ littermate female mice that were considered as control.

Both CT2A and GL261 cell lines were used for blood vessel and macrophages characterization experiments. CT-2A cell line was used for anti-CSF1, bone marrow transplant, and Vegf$^{lox/lox}$ LysMCre mice experiments.

    

Housing and all experimental animal procedures were performed in accordance with Belgian law on animal care and were approved by the Institutional Animal Care and Research Advisory Committee of the KU Leuven (P105/2012).

## Live imaging

For multiphoton excitation of endogenous fluorophores in experimental gliomas, we used a Leica SP8 *in vivo* imaging system equipped with a Chameleon Vision II titanium:sapphire laser (Coherent Inc.) and integrated with devices for intravital imaging on mice (stereotactic frame and body temperature monitoring/control). Acquisition of ROSA$^{mT/mG}$ reporter mouse was performed at 950-nm fixed wavelength. BFP signal from genetically modified tumor cells was acquired at 850-nm wavelength.

## Immunofluorescence staining

200-µm brain vibratome sections were blocked and permeabilized in TNBT (0.1 M Tris pH 7.4; NaCl 150 mM; 0.5% blocking reagent from Perkin Elmer, 0.5% Triton X-100) for 4 h at room temperature. Tissues were incubated with primary antibodies anti-Glut1 (Millipore, 1:200), anti-Glut1 (Abcam, 1:200), anti-F4/80 (Life Technologies, 1:100), anti-MHCII (Thermo Scientific, 1:100), anti-MRC1 (R&D Systems, 2 µg/ml), anti-MRC1 (Bio-Rad, 1:100), anti-CD68 (Bio-Rad, 1:100), anti-Hif1α (Thermo Scientific, 1:100), anti-CD31 (Abcam, 1:100), anti-CD3 (Abcam, 1:100), anti-CD19 (Cell Signaling Technology, 1:100), anti-CD11c (AbD Serotec, 1:100), anti-NK1-1 (Biolegend, 1:100), LY6G (Biolegend, 1:100), anti-Ki67 (Abcam, 1:100), anti-active Caspase3 (Abcam, 1:200), or anti-phospho-H2AX-S139 (Cell Signaling Technology, 1:100) diluted in TNB Triton buffer overnight at 4°C, washed in TNT buffer (0.1 M Tris pH 7.4; NaCl 150 mM, 0.5% Triton X-100) and incubated with appropriate secondary antibody (coupled with Alexa 488/555/647, Life Technologies, 1:200) diluted in TNB Triton buffer overnight at 4°C. Tissues were washed and mounted on slides in fluorescent mounting medium (Dako). Images were acquired using a Leica TCS SP8 confocal microscope.

For human samples, fresh tissues were rinsed in PBS and fixed overnight in 4% PFA at 4°C. Samples were washed in PBS and sectioned (200 µm) at the vibratome. Sections were subjected to a first step of permeabilization in −20°C cold methanol for 1 h at −20°C. Samples were washed in PBS, and the standard staining protocol was then applied starting with blocking and permeabilization in TNBT buffer.

## *In situ* hybridization

Probes for ISH were made by PCR amplification and verified by sequencing. PCR products were cloned into Zero Blunt TOPO (Life Technologies) according to the instructions of the manufacturer.

*In situ* hybridization was realized on 5-week growth mouse glioma sample 4% PFA fixed 15-µm cryosections. Sections were dried and fixed in PFA 4% for 20 min at room temperature. After washes, the samples were treated with 5 µg/ml proteinase K/PBST for 11 min and post-fixed for 15 min in 4% PFA. After washing in PBST, samples were incubated with hybridization mix (50% formamide, 1.3× SSC, 5 mM EDTA, 50 mg/ml yeast RNA, 0.2% Tween-20,

100 mg/ml heparin, 10% CHAPS) for 1 h at 70°C and then overnight in hybridization mix containing the probe at 70°C. The following day, samples were extensively washed with hybridization mix and MABT (100 mM maleic acid, 150 mM NaCl, 1% Tween-20, pH 7.5) before being blocked for 2 h in MABT/2% blocking reagent (Boehringer Mannheim)/20% goat serum. The samples were then incubated overnight at room temperature with blocking buffer containing anti-DIG-AP antibody (Roche). Next, the samples were washed in MABT for 48 h, rinsed in fresh NTMT (100 mM NaCl, 100 mM Tris–HCl pH 9.5, 50 mM MgCl2, 10% Tween-20), and developed with NTMT/0.33 mg/ml nitroblue tetrazolium (NBT), 0.05 mg/ml 5-bromo-4-chloro-3-indolyl-phosphate (BCIP) at 37°C.

## Soluble Flt-1 binding assay

Freshly sectioned 200-µm brain slices were blocked and permeabilized in TNBT (0.1 M Tris pH 7.4; NaCl 150 mM; 0.5% blocking reagent from Perkin Elmer, 0.5% Triton X-100) for 2 h at room temperature. Tissues were incubated with 1 µg/ml recombinant mouse soluble Flt-1 FC chimera (R&D Systems) diluted in TNB Triton buffer for 2 h at room temperature. Samples were rinsed three times in TNT buffer and subjected to 4% PFA fixation for 2 min. Samples were washed in TNT buffer and incubated in Alexa647 coupled anti-human IgG secondary antibodies (Life Technologies, 1:200) diluted in TNB Triton buffer overnight at 4°C. Tissues were washed and mounted on slides in fluorescent mounting medium (Dako). Images were acquired using a Leica TCS SP8 confocal microscope.

## Glioma cell culture

CT-2A and GL261 glioma cells were cultured in DMEM (Life Technologies) supplemented with 10% FBS (Life Technologies), 1% penicillin/streptomycin (Life Technologies), and 1% glutamine (Life Technologies) until a maximum of 10 passages. Spheroids were obtained by seeding the glioma cells for 48 h on non-adherent culture dishes. Spheroids of 200–250 µm were selected for implantation.

## Vessel perfusion assay

Glioma-bearing mice from 2- to 5-week growth were anesthetized and injected intravenously with 100 µl of FITC labeled 2,000,000 MW dextran (Life Technologies). Blood vessel perfusion was visualized *in vivo* using our live imaging settings.

## Vessel permeability assay

For Miles assay, glioma-bearing mice from 2- to 5-week growth were anesthetized and injected intravenously with 100 µl 1% Evan's blue solution (Sigma). Thirty minutes after injection, mice are sacrificed and transcardially perfused with 2% PFA solution. Dissected tumors were weighed and incubated in formamide solution at 56°C for 24 h to extract the dye. The absorbance of the solution was measured with a spectrophotometer at 620–405 nm. Three mice per week were analyzed. Data are expressed as fold increase compared to 2-week glioma growth with weight normalization.

## Bone marrow transplantation

Recipient 10-week-old mice were lethally irradiated (9.5 Gy) and then intravenously injected with $10^7$ bone marrow cells from LifeAct-GFP mice 16 h later. Tumor implantation experiments were initiated 5 weeks after bone marrow reconstitution. Blood cell count was determined using a hemocytometer on peripheral blood collected by retro-orbital bleeding.

## Flow-cytometric staining of tumor-infiltrating myeloid cells

Tumor-bearing mice were anaesthetized by IP injection of 100 μl sodium pentobarbital. Both femoral arteries were opened, and the animals were perfused through the left cardiac ventricle with 50 ml cold PBS. Tumors were harvested and incubated with RPMI medium containing 10% FCS, 2.5 mg/ml collagenase D, and 5 U/ml DNase I for 30 min at 37°C. After incubation, the digested tissue was passed through a cell strainer and thoroughly washed. Cells were stained with the following monoclonal antibodies: anti-CD45 Alexa Fluor® 700 (eBioscience), anti-CD11b BV421 (BD), anti-Ly6C Alexa Fluor® 647 (AbD Serotec), anti-Ly6G FITC (BD), anti-MHCII PerCP-Cy5.5 (Biolegend), and anti-MRC1 PE (Biolegend). For the MRC1 staining, cells were fixed and permeabilized by using the LEUCOPERM™ reagents (AbD Serotec). As a control, cells were stained with the appropriate isotype control. To exclude dead cells, cells were stained with the Zombie Yellow™ dye (Biolegend). Data acquisition was performed on the BD LSRFortessa and analysis was performed with FlowJo_V10.

## MHCII-positive cell tracking

ROSA^mTmG glioma-bearing mice of 3-week growth were anesthetized and injected intravenously with 200 μl of FITC labeled anti-MHCII HLA-DR (BD Biosciences). After 6 or 24 h of antibody circulation, mice were sacrificed and transcardially perfused with 2% PFA solution. Harvested brains were then subjected to immuno-labeling.

## Macrophages extraction and qPCR analysis

Three-week growth CT-2A gliomas were harvested from six wild-type bearing mice, minced with a scalpel, and incubated in 5 ml Dulbecco's modified Eagle's medium containing 2 mg/ml collagenase I (Invitrogen) for 45 min at 37°C with shaking every 15 min followed by filtering through a 40-μm nylon mesh (BD Falcon). The cells were then centrifuged at 1,000 g for 5 min at 4°C, resuspended in buffer 1 (0.1% bovine serum albumin, 2 mM EDTA, pH 7.4, in PBS), separated equally into two tubes, and incubated with either anti-rat immunoglobulin G-coated magnetic beads (Invitrogen) precoupled with rat anti-mouse MHCII antibody (Thermo Scientific, 1 μg) or rat anti-mouse MRC1 antibody (Bio-Rad; 1 μg) for 30 min at 4°C in an over-head shaker. Beads were separated from the solution with a magnetic particle concentrator (Dynal MPC-S, Invitrogen). The beads were washed five times with buffer 1 and centrifuged for 5 min at 1,000 g, and the supernatant was removed. The purified MHCII or MRC1 macrophages were then shock-frozen in liquid nitrogen and stored at −80°C until further use.

Total RNA was isolated using the RNAeasy kit from QIAGEN.

Real-time quantitative PCR (qPCR) reactions were performed in duplicate using the MyIQ real-time PCR system (Bio-Rad). Each 25-μl reaction contained 5 ng cDNA, 12.5 μl iQ SYBR Green Supermix (Bio-Rad), 250 nM forward and reverse primers, and nuclease-free water. All qPCR primers were purchased from Qiagen (QuantiTect, Qiagen). Fold changes were calculated using the comparative CT method.

## Patient characteristics

Tumors were obtained from 29 patients after informed consent and approval of the study by the local ethical committee. Based on central review histopathology: 11 patients were diagnosed with primary glioblastoma multiforme (GBM) grade IV, eight patients with relapsed GBM grade IV, three patients with primary anaplastic oligodendroglioma grade III, one patient with primary anaplastic astrocytoma grade III, five patients with primary oligodendroglioma grade II and one patient with primary diffuse astrocytoma grade II. Patient characteristics are summarized in Table EV1.

## In vivo combination of anti-CSF1 mAb treatment and temozolomide chemotherapy

Mice were intracranially injected with GL261 tumor cells as previously described (Maes et al, 2009). Briefly, $0.5 \times 10^6$ tumor cells were injected at 2 mm lateral and 2 mm posterior from the bregma at a depth of 3 mm below the dura mater by using a stereotactic frame (Kopf Instruments). Stereotactic inoculation was performed under sterile conditions.

Temozolomide (Temodal 20-mg capsules, Schering-Plough) was delivered to the mice by gavage. In order to improve the solubility of TMZ, an equal dose of L-histidine (Calbiochem/Merck) was added. For each capsule that contained 20 mg TMZ, 20 mg histidine was added and dissolved in a phosphate buffer. To prepare the buffer, 2.72 g of potassium dihydrogen phosphate was added in 1,000 ml sterile water and pH 5 was achieved by adding 1N potassium hydroxide. The TMZ–histidine–phosphate buffer mixture was then sonicated for 1 h, and the tube was shaken every 20 min to avoid deposit (Branson Ultrasonics). Mice received a dose of 40 mg/kg in 0.2 ml at days 7, 11, and 15 after tumor inoculation.

As a control of temozolomide side effect on immune cell population, the same administration was performed on 12-week-old C57BL/6J mice separated into four groups: untreated without glioma implantation ($n = 3$); untreated with glioma implantation ($n = 3$); treated without glioma implantation ($n = 3$); and treated with glioma implantation ($n = 3$). Three days after the last administration, mice were sacrificed and femur and tibia were isolated. From these bones, bone marrow was flushed using PBS followed by erythrocyte lysis. Cells were stained for zombie near infrared (Biolegend) as a viability dye followed by surface staining for T-cell population markers and general myeloid markers using the following monoclonal antibodies: PerCP-Cy5.5-conjugated anti-CD4 (eBioscience), BV605-conjugated anti-CD11b (Biolegend), and APC-conjugated anti-CD11c (eBioscience). Flow cytometry was performed using a LSR Fortessa Analyzer (BD Biosciences, Erembodegem, Belgium) and analyzed using FlowJo software (Tree Star, Ashland, OR, USA).

Anti-CSF1 mAb was diluted in saline (NaCl 0.9%, Braun) and intraperitoneal administered at a dose of 0.5 mg at days 4, 8, 12, and 16 after tumor inoculation.

The endpoint of the experiment was defined as three times the median survival of the control group.

**Melanoma inoculation procedure**

Male mice at 10–12 weeks of age (C57Bl/6J) were used. Anti-CSF1 mAb treatment was started 2 days after subcutaneous injection of $5 \times 10^6$ B16F10 cells. Three doses of 0.5 mg of anti-CSF1 mAb (BioXCell) were given by intraperitoneal (i.p.) injection at days 2, 6, and 10. Tumor size was measured after the animal was sacrificed.

**Statistical analysis and quantification**

For continuous variables, data are presented as mean ± SD or medians (interquartiles) as appropriate.

Between-group comparisons used the Mann–Whitney $U$-test or $t$-test depending on the sample size for continuous variables.

In cases when more than two groups were compared, one-way ANOVA test was performed, followed by Turkey's multiple comparison test, and results were considered significantly different if $P < 0.05$.

For correlation analysis, Spearman's correlation test was performed.

For survival experiment, log-rank (Mantel–Cox) tests and multiple comparison test were performed.

A two-tailed value of $P < 0.05$ was considered statistically significant.

All the analyses were performed using Prism 6.0 software (GraphPad).

For mice *in vivo* imaging quantification, three fields per animal were pictured in the tumor center and blood vessel caliber, branching, vessel perfusion, and distance of macrophages to blood vessel were analyzed using Fiji software.

For mice *ex vivo* and human sample imaging quantification, five fields per individual were pictured in the tumor center (or tumor periphery when mentioned) and vessel caliber, vessel branching, number of macrophages and resident macrophages, overlapping stainings, macrophages proliferation, hypoxic area, and tumor double-strand DNA damages were quantified using Fiji software.

Expanded View for this article is available online.

**Acknowledgments**

The research was supported by the Belgian Cancer Foundation (Stichting Tegen Kanker, grant 2012-181), the European Research Council Consolidator Grant REshape (311719), and a Hercules type 2 grant (Herculesstichting: AKUL11033). T.M. is funded by a EMBO Long-Term Fellowship. S.V. is supported by BBTS (Belgian Brain Tumor support) and HMRF (Herman Memorial Research Foundation). T.V was funded by a FWO-V Postdoctoral Fellowship. MVW is holder of an IWT-bursary. M.M is supported by an ERC Starting Grant (OxyMO). K.M. is supported by a JSPS Postdoctoral Fellowship. GL261 and CT2A cells were kindly provided by Dr. Till Acker (Institute of Neuropathology, University of Giessen, Germany) and Dr. Thomas N. Seyfried (Biology department, Boston College, USA), respectively.

**The paper explained**

**Problem**
Central nervous system associated tumors require blood vessel supply to sustain their growth. Glioma, the most predominant form of primary brain tumors, is characterized by blood vessel abnormalities that impede efficient drug delivery.

**Results**
To understand the dynamics and underlying principles of blood vessel patterning during glioma growth, we performed *in vivo* longitudinal imaging using multiphoton microscopy of mouse glioma with detailed cellular resolution. We discovered that blood vessel patterning defects emerge as a late characteristic of glioma growth and are associated with a particular switch in the phenotype and location of innate immune cell (macrophages) within the tumor microenvironment. We show that these stromal cells recruited as M1 cytotoxic macrophages to fight tumor cells switch in the tumor microenvironment to a M2 tumor supportive phenotype and aggregate around blood vessels where they produce the key endothelial cell growth factor VEGF-A. Timelapse imaging illustrated that endothelial cells under these conditions fail to sprout and branch, and instead show synchronous but ineffective protrusive activity driving vessel growth in size rather than branching. Depletion of macrophages using anti-CSF1 strategy or genetic depletion of macrophage-derived VEGF-A production normalized vascular patterning and restored vessel functionality, substantially improving chemotherapeutic agent efficacy and survival in glioma-bearing mice.

**Impact**
From a clinical perspective, the actual phenotypic progression of the vascular network and the immune cell recruitment appear highly similar in human samples, providing important new insights into the longitudinal development of tumor vessel density and functionality.

**Author contributions**

HG supervised the project. HG and TM designed experiments. TM, CB, TV, FS, MVW, L-KP, MB, JB, KM, and PBG performed experiments. TM, CB, TV, L-KP, JD, and RS analyzed data. HG, TM, CS, MM, and SDV interpreted the data. HG and TM wrote the manuscript.

**Conflict of interest**

The authors declare that they have no conflict of interest.

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
