## [Review Process File · EMBO Molecular Medicine]

Dynamic stroma reorganization drives blood vessel dysmorphia during glioma growth

Thomas Mathivet, Claire Bouleti, Matthias Van Woensel, Fabio Stanchi, Tina Verschuere, Li-Kun Phng, Joost Dejaegher, Marly Balcer, Ken Matsumoto, Petya B Georgieva, Jochen Belmans, Raf Scot, Christian Stockmann, Massimiliano Mazzone, Steven De Vleeschouwer and Holger Gerhardt

*Corresponding authors: Thomas Mathivet and Holger Gerhardt
Vesalius Research Center, VIB, KU Leuven*

Review timeline:

Submission date:	08 December 2016
Editorial Decision:	11 January 2017
Revision received:	30 July 2017
Editorial Decision:	29 August 2017
Revision received:	12 September 2017
Accepted:	13 September 2017

Transaction Report:

Editor: Roberto Buccione

1st Editorial Decision

11 January 2017

Thank you for the submission of your manuscript to EMBO Molecular Medicine and please accept our apologies for the unusual delay, due also to the concomitant holiday season.

We have now heard back from the three Reviewers whom we asked to evaluate your manuscript.

All three reviewers find the manuscript of interest but also express several fundamental concerns. Reviewer 1 and 2 are more reserved and point to complementary and in part overlapping issues, among which 1) lack of mechanisms explaining how (and if) M2 macrophages induce vessel dysmorphia including the uncertain origin of VEGF, 2) unclear specificity of the observations for glioma, 3) whether myeloid cells are involved and 4) unconvincing chemotherapy & CSF1 data. The same reviewers, together with Reviewer 3, also ask for additional quantification and detail, re-writing of the manuscript, and mention the poor overall quality of data presentation including statistical analysis, and poor referencing to previous work.

Overall, I think that there is clear appreciation for the inherent experimental challenges and the clever approach, but also that there needs to be a significant upgrade in terms of data consolidation and mechanistic insight to consider publication in EMBO Molecular Medicine.

Finally, I should mention that during our Reviewer cross-commenting exercise, there emerged an agreement on the need to address the above issues but also a consensus (including with myself) that perhaps the analysis of the mechanisms supporting the M1/M2 switch and the isolated M2

macrophage injection experiment are further reaching and/or unlikely to provide data crucial for this work.

In conclusion, while publication of the paper cannot be considered at this stage, given the potential interest of your findings and after internal discussion, we have decided to give you the opportunity to address the criticisms. Please consider that the concerns raised are of great importance for us as they impinge on the most interesting potential messages of the manuscript.

We are thus prepared to consider a substantially revised submission, with the understanding that the Reviewers' concerns must be addressed with additional experimental data where appropriate, save for the items mentioned above and that acceptance of the manuscript will entail a second round of review.

Please note that it is EMBO Molecular Medicine policy to allow a single round of revision only and that, therefore, acceptance or rejection of the manuscript will depend on the completeness of your responses included in the next, final version of the manuscript.

EMBO Molecular Medicine now requires a complete author checklist (<http://embomolmed.embopress.org/authorguide#editorial3>) to be submitted with all revised manuscripts. Provision of the author checklist is mandatory at revision stage; The checklist is designed to enhance and standardize reporting of key information in research papers and to support reanalysis and repetition of experiments by the community. The list covers key information for figure panels and captions and focuses on statistics, the reporting of reagents, animal models and human subject-derived data, as well as guidance to optimise data accessibility.

As you know, EMBO Molecular Medicine has a "scooping protection" policy, whereby similar findings that are published by others during review or revision are not a criterion for rejection. However, I do ask you to get in touch with us after three months if you have not completed your revision, to update us on the status. Please also contact us as soon as possible if similar work is published elsewhere.

Finally, we now mandate that all corresponding authors list an ORCID digital identifier. You may acquire one through our web platform upon submission and the procedure takes <90 seconds to complete. We also encourage co-authors to supply an ORCID identifier, which will be linked to their name for unambiguous name identification.

I look forward to seeing a revised form of your manuscript as soon as possible.

***** Reviewer's comments *****

Referee #1 (Comments on Novelty/Model System):

see point 9 of my comments

Referee #1 (Remarks):

This is an elegant dynamic study suggesting that the chaotic vasculature characterizing experimental and human gliomas results from the combination of two sequential steps. The first is characterized by the appearance and increased of capillaries in the growing tumors through the mechanism of sprouting angiogenesis; the second is governed by M2 macrophages which induce the vessel changes of the shape. The authors sustain this conclusions by combining intra-vital live imaging in genetically modified mouse models with a treatment to deplete macrophages. Furthermore they show that the depletion of macrophages by an anti-CSF1 Ab improves the therapeutic effect of temozolomide in an experimental glioma suggesting that CSF1 depletion improves drug delivery.

Generally, it is well a planned work containing relevant pre-clinical findings that could be exploited in therapeutic strategies. However, the present version of the MS does not contain any mechanism explaining the M1/M2 switch and how M2 macrophages induce vessel dysmorphia.

CRITICISMS

1. The data showing that M2 phenotype appears concomitantly with dysmorphic vessels and macrophage depletion reverts this alteration suggest but do not demonstrate that M2 macrophages drive these vascular modifications (Fig 6,7). The following questions are open and their responses could reinforce the authors' hypothesis. Do M2 macrophages (isolated from late phase of glioma progression (5 wks) or in vitro differentiated) injected in the early phase of the glioma progression (2 wks) anticipate vascular dysmorphia? Another important control is the evaluation of M2 effect on normal capillaries. This aspect can be easily studied by injecting M2 macrophages in normal brain.
2. Vascular dysmorphia is a common feature of solid tumors and it is not strictly connected with glioma behaviour. Is the role of M2 macrophages specific for gliomas or a general properties in other solid tumors. I think that this group can easily reply to this comment by repeating the above suggested experiment in another tumor model.
3. By combining in situ analysis and macrophages derived from Vegfa1/fl:LysMCre mice, the authors support that VEGFA produced by macrophages surrounding vessels mediates the dysmorphic phenotype. However this strategy does not take into account other sources of VEGFA. The authors have to show the effect of total VEGF removal in the late phase of their experimental protocol.
4. The hypothesis that VEGF released by M2- macrophages is mainly based on the experiments shown in Fig s8. This picture is not so informative. A deep quantitative analysis of the VEGF expression near vessels is required. I suggest to quantify VEGF in at least 3 different areas located at different distances from the vessels and show a real enrichment in the closest area.
5. MDSCs have important role in tumor progression and in tumor angiogenesis. Hypoxia, which is a hallmark of gliomas, promotes MDSC recruitment. Can the author exclude a role of this myeloid subtype in the described phenotype? I suggest to analyze the presence of these cells in the different experimental conditions proposed in the paper.
6. The conclusion (line 363) derived from experiments shown in Figure 8 is not correct. To state that anti-CSF1 "substantially improved delivery of temozolomide", the author have to measure the compound in the tissue. Actually the authors cannot exclude different pharmacodynamics effects mediated by the dual therapy (see for example DOI 10.15252/emmm.201505774)
7. In solid tumors as well as in gliomas hypoxia is strictly correlated with vascular dynamics. I'm a little bit surprised that the authors do not describe the changes of hypoxic areas along their experimental windows (2 and 5 wks) and they are modified by the modulation of vascular shape and M2 recruitment. I think this point has to be carefully addressed.
8. I'm aware that the description of the mechanisms supporting the M1/M2 switch are out the aim of the work, but I invite the authors to discuss this point
9. In many experiments I suggest deeper quantification of the phenotype described showing not only IF pictures but also the quantification of the data shown with appropriate statistical analysis

Referee #2 (Comments on Novelty/Model System):

This manuscript is interesting, and the data presented are very relevant to the therapeutic targeting of myeloid cells in solid tumors, currently the subject of several undergoing clinical trials. The authors show that there is a switch from a more ordered blood vessel hierarchy to dysmorphic vasculature during glioma progression, that can be prevented by the depletion of macrophages, and that macrophage-produced VEGF seems to be at least partially responsible for this change. Furthermore, CSF-1 inhibition also improves the efficacy of chemotherapy in this model, despite somewhat surprisingly causing accelerated tumor growth on its own.

Referee #2 (Remarks):

This manuscript is interesting, and the data presented are very relevant to the therapeutic targeting of

myeloid cells in solid tumors, currently the subject of several undergoing clinical trials. The authors show that there is a switch from a more ordered blood vessel hierarchy to dysmorphic vasculature during glioma progression, that can be prevented by the depletion of macrophages, and that macrophage-produced VEGF seems to be at least partially responsible for this change. Furthermore, CSF-1 inhibition also improves the efficacy of chemotherapy in this model, despite somewhat surprisingly causing accelerated tumor growth on its own.

The use of intravital imaging is ambitious, and allows longitudinal study and observation of real-time dynamics, but some of the data needs to be better presented and more complete to justify the conclusions. The analysis of macrophage infiltration and vascular characteristics does not extend to the chemotherapy study, weakening the conclusions. The discussion would benefit from editing for better focus and readability, and this work needs to be better put in the context of the extensive existing data from related experiments.

The detailed comments are below

1. Intravital imaging

The authors have used intravital imaging, which has the advantage of the possibility to image the same tumor (even the same location) over time in more and less advanced tumor stages. However, it is not clearly stated in the methods or figures whether the same mice were in fact used for characterization of early and late tumors, although presumably this was the case. Are the early and late stage still images presented in the figures from the same tumor, or just representative? (Also, it is not always clear if quantifications are done from intravital images or sections.)

The disadvantages of intravital imaging include the extremely limited area of the tumor that can be visualized, limited to the very top of the tumor. Also, often a small number of animals is used due to the slow and tricky method. Were parameters such as vessel architecture or macrophage number & location similar in deeper regions of the tumor?

Where intravital imaging is indispensable in this manuscript is the description of vascular sprouting dynamics in early and late tumors, but this part seems somewhat incomplete. It is not trivial to tell the difference between videos 1 and 4, when the scale of the vessels in the field of view is so different, and there are hardly any explanatory comments in the legends, not to mention arrows/other annotations in the videos. Maybe zooming in on Video 1 would help? Also, to make a point about sprouting being different in early and late tumors, this effect needs to be quantified in some way. Was sprouting behavior normalized with anti-CSF-1?

In the current format, the role intravital imaging as a method is over-emphasized in the last two chapters of the introduction.

2. Chemotherapy & aCSF-1 experiment

On line 295, a reference to the "classic" cell suspension injection model is missing, and/or an explanation for why this was chosen instead of the spheroids used for all other experiments.

This experiment seems somewhat incomplete. Did chemotherapy recruit additional bone-marrow derived cells into the tumor, and were these macrophages or perhaps monocytes/neutrophils? Did aCSF-1 deplete these TMZ-recruited cells? What were the effects of chemotherapy and combination therapy on vasculature? How about tumor volume? Hypoxia? Sprouting dynamics?

It is stated in the manuscript that "cell death" was more homogenous with combination therapy than with chemotherapy alone. This needs to be quantified. An apoptosis marker (activated Caspase 3?) would be a better indicator of cell death than the DNA damage marker pH2AX. Can the cell death observed be correlated to density/"normality" of the vasculature?

It is not clear from the methods if multiple comparisons have been corrected for in the p-values for the survival curves, and how?

Inhibition of CSF-1R/macrophage depletion has been shown to improve responses to chemotherapy

or radiotherapy in tumor models in several studies (at least DeNardo et al, *Cancer Discov* 2011; Mitchem et al, *Cancer Res* 2013; Xu et al, *J Urol* 2014; Shiao et al, *Cancer Immunol Res* 2015). Also Hughes et al, *Cancer Res* 2015 is of interest. This existing literature should be referred to and discussed in this manuscript.

3. References and discussion of previous research

The manuscript includes several outdated references, which omit important more recent data. Ref 1 is textbook knowledge and can be left out. Outdated references include Ref4 (replace with e.g. Carmeliet & Jain 2011), Ref 5 (e.g. Noy & Pollard 2014, Qian & Pollard 2010), Ref 7 (replace with e.g. Zumsteg & Christofori 2009, or add another reference such as Coffelt et al 2009, or Qian & Pollard 2010, which also has a full chapter on tumor macrophages and angiogenesis)

Discussion lines 344-346: For completeness, add Cotechini et al *Cancer J* 2015 review; among other data it has a list of all myeloid-targeted therapeutics currently in clinical trials.

The discussion should include a slightly more extensive treatment of how this manuscript fits into the context of previous data on macrophage depletion, CSF-1/CSF-1R inhibition and angiogenesis. A large body of data proves that macrophages stimulate tumor angiogenesis in a large variety of models, especially in breast cancer models, and most often macrophage depletion and accompanying reduced angiogenesis has led to attenuated tumor growth; myeloid cells are also involved in intra- and extravasation and support invasion and metastasis. In gliomas, myeloid cell depletion (De Palma et al, *Cancer Cell* 2005; Zhai et al, *Glia* 2010) or CSF-1R inhibition (Pyonteck et al 2013) also had anti-tumor effects, although acceleration of tumor growth, as in this manuscript, has also been shown before (Galarneau et al, *Cancer Res* 2011; Stockmann et al 2008).

4. Miscellaneous

Results section, lines 231-237: some/all could be moved to the discussion.

Macrophage depletion was only 50% - discuss if a more complete depletion (with anti-CSF-1R?) would have given different results? Were the remaining macrophages M1 or M2 polarized?

There is no description in the methods of how tumor volumes were measured.

Glut1 has been used as a pan-endothelial marker in the human glioma samples. Is there no reason to believe that its expression is downregulated in abnormal vessels in high-grade gliomas?

The slight increase in neutrophil recruitment due to CSF-1 inhibition shown in Supplementary Figure 11 should be more directly mentioned in the results and discussed.

Discussion, lines 350-355 are a bit confusing. If IL-34 is not present at all in glioma ("controversial"), CSF-1R inhibition would presumably not have very different effects from CSF-1 inhibition? In any case, inhibition of both ligands (and possible ligand-independent effects?) in a cancer setting is not necessarily a disadvantage.

Methods, line 414, title should be Immunofluorescent staining only?

Methods, line 478, use more scientific language.

The manuscript is well understandable, but grammar checking by a native English speaker would improve it.

Referee #3 (Remarks):

This is a very interesting contribution from a laboratory interested in blood vessel patterning in developing organs. In the present work, the authors investigate tumor angiogenesis, with the goal to establish similarities and differences between normal and tumor vessel patterning. They use sophisticated multiphoton imaging of glioma cells implanted into mice, and show that while the

tumor vessels are initially similar to normal vessels with individual tip and stalk cells, over time, they progressively become enlarged and dysfunctional. This abnormalization correlates with recruitment of bone-marrow derived macrophages and a switch from a M1 to a M2 phenotype. Analysis of human glioma samples reveals that both vessel enlargement and M2 macrophage recruitment are remarkably well conserved in human high grade glioblastoma. Blocking macrophage recruitment using anti-CSF1 antibodies prevents blood vessel abnormalization, while recombinant CSF1 treatment of early stage tumors enlarges tumor vessels. M2 macrophages produce Vegf, and the authors show that genetic blockade of macrophage Vegf production deletion prevents vessel abnormalization. Tumors with better blood vessels grow faster, and are more sensitive to Temozolimid, providing an approach to enhance delivery of cytotoxic agents via improved vasculature. Overall, I find that the work is very well executed, the data are clear and the message is highly interesting to the wide readership of Embo Mol Med. I am in favor of publication but have a number of minor issues the authors should address to improve the presentation of the data.

Specific comments:

1. The illustration needs a complete overhaul. All the legends of all the graphs are way too small and impossible to read on a printout. Please increase font size.
2. Statistic methods should be included in the figure legends.
3. Fig.3. Legend title doesn't make sense. I guess switch is meant rather than in situ.
4. The blue macrophage staining is hard to see. Could it be changed to a different color to be easier to see?
5. The Introduction is very short and does not highlight the novelty of the paper. In my mind, there are three novel aspects that could be mentioned. First, the study directly demonstrates vessel abnormalization by longitudinal imaging. I am not aware of other studies using this methodology to image tumor vessel development, if there are, the authors should cite them! Second, while it is known that macrophages affect tumor progression, this study shows that macrophages affect the vasculature, which again to my knowledge has not been reported before. Third, they show that macrophages induce alterations in Vegf gradients, and that it is the change in the gradient of this factor, rather than the presence of a hypothetical tumor angiogenesis factor that leads to the chaotic nature of the intratumor vasculature.

1st Revision - authors' response

30 July 2017

Referee 1

Referee #1 (Comments on Novelty/Model System):

see point 9 of my comments

Referee #1 (Remarks):

This is an elegant dynamic study suggesting that the chaotic vasculature characterizing experimental and human gliomas results from the combination of two sequential steps. The first is characterized by the appearance and increased of capillaries in the growing tumors through the mechanism of sprouting angiogenesis; the second is governed by M2 macrophages which induce the vessel changes of the shape. The authors sustain this conclusions by combining intra-vital live imaging in genetically modified mouse models with a treatment to deplete macrophages. Furthermore they show that the depletion of macrophages by an anti-CSF1 Ab improves the therapeutic effect of temozolomide in an experimental glioma suggesting that CSF1 depletion improves drug delivery.

Generally, it is well a planned work containing relevant pre-clinical findings that could be exploited in therapeutic strategies. However, the present version of the MS does not contain any mechanism explaining the M1/M2 switch and how M2 macrophages induce vessel dysmorphia.

We thank the referee for his positive feedback on our work. We also appreciate and share the desire to understand how M1 macrophages switch to M2 and how the latter induce vessel dysmorphia. It has long been debated whether this switch represents de novo recruitment of distinct populations or entails in situ repolarization of macrophages. To our knowledge this question is still unresolved, however, we provide first data using an in vivo pulse of MHCII antibody staining and timed “chase” by performing post-fixation staining for MRC1. Whereas simultaneous double staining shows no overlap, thus indicating distinct marker distribution and populations, the 24h chase identifies significant double positive populations. This shows that cells that had earlier expressed high levels of MHCII, turned MRC1 positive. We feel that the identification of the key drivers of this switch on the molecular level in the in vivo setting will need to be addressed in future studies. A cytokine profile that can achieve this in vitro is well established, but the in vivo identification of the correct ones and where they are produced will be a complete study in its own right.

Regarding the mechanism of vessel dysmorphia, we present evidence for an important role of M2 derived VEGF-A. In the substantially revised manuscript, we now include qPCR analysis of isolated macrophage populations showing a highly selective co-expression of the M2 marker MRC1 and VEGF-A (new figure 7a).

Together with the in situ hybridization, these data strongly suggest that VEGF-A production by M2 macrophages and the clustering of this VEGF source around vessels right at the stage when dysmorphia occurs, is at least part of the mechanism. The observation that genetic deletion of VEGF-A only in myeloid cells prevents much of the dysmorphia represents further mechanistic evidence. We now performed additional VEGF sequestration using treatment with sflt1, and find that both inhibiting macrophages by anti-CSF1 or inhibiting VEGF by sflt1 produces similar restoration of vessel patterning (new supplementary figure 12). Interestingly however, genetic deletion of VEGF-A and sequestration of total VEGF-A show differential effects on tumor growth, suggesting that myeloid VEGF is driving vessel dysmorphia, whereas total VEGF has additional effects.

CRITICISMS

1. The data showing that M2 phenotype appears concomitantly with dysmorphic vessels and macrophage depletion reverts this alteration suggest but do not demonstrate that M2 macrophages drive these vascular modifications (Fig 6,7). The following questions are open and their responses could reinforce the authors' hypothesis. Do M2 macrophages (isolated from late phase of glioma progression (5 wks) or in vitro differentiated) injected in the early phase of the glioma progression (2 wks) anticipate vascular dysmorphia? Another important control is the evaluation of M2 effect on normal capillaries. This aspect can be easily studied by injecting M2 macrophages in normal brain.

We thank the reviewer for the interesting approach proposed here, which could potentially strengthen our message. Nevertheless, as mentioned by the editor in the decision letter, the potential experiments of M2 macrophages re-implantation might not bring satisfactory answers (“perhaps the analysis of the mechanisms supporting the M1/M2 switch and the isolated M2 macrophage injection experiment are further reaching and/or unlikely to provide data crucial for this work”).

2. Vascular dysmorphia is a common feature of solid tumors and it is not strictly connected with glioma behaviour. Is the role of M2 macrophages specific for gliomas or a general properties in other solid tumors. I think that this group can easily reply to this comment by repeating the above suggested experiment in another tumor model.

This is a relevant question. In order to assess whether our results were specific to the glioma settings or more generally applicable to solid tumor, we performed B16 melanoma injection combined with anti-CSF1 Ab treatment, new data now presented in Supplementary Figure 13 (n=5 mice per group). The similarities between the glioma and melanoma models suggest this is a more general mechanism and the link between macrophages and vessel dysmorphia may apply to many solid tumors and their progression.

3. By combining in situ analysis and macrophages derived from Vegf^{fl/fl}:LysM^{Cre} mice, the authors support that VEGFA produced by macrophages surrounding vessels mediates the dysmorphic phenotype. However this strategy does not take into account other sources of VEGFA.

The authors have to show the effect of total VEGF removal in the late phase of their experimental protocol.

To answer this important question, we injected sFlt1 as a VEGF-A trap i.p. every other day starting from one week post glioma implantation (n=5 mice per group). The results are presented in Supplementary figure 12 and demonstrate that VEGF depletion induces a vascular normalization in late stage glioma growth.

4. The hypothesis that VEGF released by M2- macrophages is mainly based on the experiments shown in Fig s8. This picture is not so informative. A deep quantitative analysis of the VEGF expression near vessels is required. I suggest to quantify VEGF in at least 3 different areas located at different distances from the vessels and show a real enrichment in the closest area.

We appreciate this practical suggestion and have now performed in depth quantitation accordingly. We quantified VEGF production (based on signal intensity measurements) in relation to blood vessel distance (3 groups: <50µm; 50 to 150µm; >150µm). The data illustrate in late stage glioma a high detection of VEGF in the close vicinity of the vessels (<50µm) corresponding to the zone of predominant M2 macrophage location, and also far away from blood vessels (>150µm), which could correspond to hypoxic tumor cells. The results are presented in Supplementary Figure 8C.

5. MDSCs have important role in tumor progression and in tumor angiogenesis. Hypoxia, which is a hallmark of gliomas, promotes MDSC recruitment. Can the author exclude a role of this myeloid subtype in the described phenotype? I suggest to analyze the presence of these cells in the different experimental conditions proposed in the paper.

To check the involvement of MDSCs in our described phenotype, we performed Ly-6C/G and CD11b co-staining. Interestingly, we failed to detect any differences in control versus anti-CSF1 Ab treatments. Given that anti-CSF1 Ab treatment led to vessel normalization, but had no effect on MDSC numbers, a role for MDSC in the vascular phenotype appears unlikely. The results are presented in Supplementary Figure 11B and D.

6. The conclusion (line 363) derived from experiments shown in Figure 8 is not correct. To state that anti-CSF1 "substantially improved delivery of temozolomide", the author have to measure the compound in the tissue. Actually the authors cannot exclude different pharmacodynamics effects mediated by the dual therapy (see for example DOI 10.15252/emmm.201505774)

We agree that firm conclusions on drug delivery would require direct measurements of drug distribution. As we cannot formally exclude altered pharmacodynamics we toned down the conclusion to "suggesting an improved delivery of temozolomide".

7. In solid tumors as well as in gliomas hypoxia is strictly correlated with vascular dynamics. I'm a little bit surprised that the authors do not describe the changes of hypoxic areas along their experimental windows (2 and 5 wks) and they are modified by the modulation of vascular shape and M2 recruitment. I think this point has to be carefully addressed.

In Supplementary Figure 9 we show the expansion of hypoxic areas during glioma growth using glut1 staining. We observed similar results with Hif1a staining.

8. I'm aware that the description of the mechanisms supporting the M1/M2 switch are out the aim of the work, but I invite the authors to discuss this point

We share the referee's desire to understand this switch mechanistically. Many groups are working on this topic, studying the signaling mechanisms, mainly in vitro. Current literature suggests it is cytokine driven, and maybe p38 mediated. However details are lacking and there is no real consensus to our knowledge. Our work begins to address an important mechanistic point, ie the question whether these are distinct populations that are recruited differentially, or rather switch in situ. This part we have discussed, but feel any further mechanistic discussion without further data would be too speculative at this point.

9. In many experiments I suggest deeper quantification of the phenotype described showing not only IF pictures but also the quantification of the data shown with appropriate statistical analysis

We agree and have now performed numerous extra quantifications to investigate the phenotype and treatment effects in the revised manuscript.

Referee 2

Referee #2 (Comments on Novelty/Model System):

This manuscript is interesting, and the data presented are very relevant to the therapeutic targeting of myeloid cells in solid tumors, currently the subject of several undergoing clinical trials. The authors show that there is a switch from a more ordered blood vessel hierarchy to dysmorphic vasculature during glioma progression, that can be prevented by the depletion of macrophages, and that macrophage-produced VEGF seems to be at least partially responsible for this change. Furthermore, CSF-1 inhibition also improves the efficacy of chemotherapy in this model, despite somewhat surprisingly causing accelerated tumor growth on its own.

Referee #2 (Remarks):

This manuscript is interesting, and the data presented are very relevant to the therapeutic targeting of myeloid cells in solid tumors, currently the subject of several undergoing clinical trials. The authors show that there is a switch from a more ordered blood vessel hierarchy to dysmorphic vasculature during glioma progression, that can be prevented by the depletion of macrophages, and that macrophage-produced VEGF seems to be at least partially responsible for this change. Furthermore, CSF-1 inhibition also improves the efficacy of chemotherapy in this model, despite somewhat surprisingly causing accelerated tumor growth on its own.

The use of intravital imaging is ambitious, and allows longitudinal study and observation of real-time dynamics, but some of the data needs to be better presented and more complete to justify the conclusions. The analysis of macrophage infiltration and vascular characteristics does not extend to the chemotherapy study, weakening the conclusions. The discussion would benefit from editing for better focus and readability, and this work needs to be better put in the context of the extensive existing data from related experiments.

We thank the reviewer for the positive appreciation of our work.

The detailed comments are below

1. Intravital imaging

The authors have used intravital imaging, which has the advantage of the possibility to image the same tumor (even the same location) over time in more and less advanced tumor stages. However, it is not clearly stated in the methods or figures whether the same mice were in fact used for characterization of early and late tumors, although presumably this was the case. Are the early and late stage still images presented in the figures from the same tumor, or just representative? (Also, it is not always clear if quantifications are done from intravital images or sections.)

Yes, the early and late stage time points are taken in the same area of the same animal and this is now mentioned in the figure legend: "Representative images of two-photon live imaging of the same glioma area of the same mouse on 2 and 5 weeks growth glioma (BFP positive) implanted in ROSA^{mT/mG}::*Pdgfb-iCre* mouse". All the quantifications matching with in vivo acquired pictures are from in vivo data.

The disadvantages of intravital imaging include the extremely limited area of the tumor that can be visualized, limited to the very top of the tumor. Also, often a small number of animals is used due to the slow and tricky method. Were parameters such as vessel architecture or macrophage number & location similar in deeper regions of the tumor?

We thank the reviewer for this remark. It is true that our *in vivo* imaging is limiting us to a visualization of no deeper than 800 μ m. To confirm the relevance of our intravital imaging results, we performed the same analysis in additional animals that were sacrificed at all the major time points and labeled post tissue fixation. To ensure that our image data were representative of the full tumor, we performed mosaic imaging from the healthy margin of the tumor to the window/skull edge and performed quantification on these data.

Where intravital imaging is indispensable in this manuscript is the description of vascular sprouting dynamics in early and late tumors, but this part seems somewhat incomplete. It is not trivial to tell the difference between videos 1 and 4, when the scale of the vessels in the field of view is so different, and there are hardly any explanatory comments in the legends, not to mention arrows/other annotations in the videos. Maybe zooming in on Video 1 would help? Also, to make a point about sprouting being different in early and late tumors, this effect needs to be quantified in some way. Was sprouting behavior normalized with anti-CSF-1?

The scales of the movies 1 and 4 are very similar and have now been added to the movies. Arrows to point ectopic sprouts have now been added and a time indication. Unfortunately we cannot directly prove that sprouting behavior was normalized with CSF-1. The fixed sample and still images would strongly suggest this is the case, but we were not able to provide new live-imaging data on the CSF1 treated glioma samples. The reason being that the first author has moved labs in the meantime and we could not line up a whole new treatment series in time. In particular as we felt that demonstrating full normalization of behavior based on time-lapse movies would require many such movies and quantitative assessment. This is not straightforward to achieve. Therefore, we must state this can only be addressed in sufficient detail in future work.

In the current format, the role intravital imaging as a method is over-emphasized in the last two chapters of the introduction.

We appreciate this concern. The manuscript contains both live imaging and fixed tissue data. However, the key aspect of longitudinal imaging at the cellular level is what has provided the insight into the mechanism of vessel dysmorphia. Only thanks to live imaging and longitudinal study did we identify the correlation between blood vessel dysmorphia and the massive myeloid cell invasion of the tumor. We have however tuned down the emphasis in the introduction as requested.

2. Chemotherapy & aCSF-1 experiment

On line 295, a reference to the "classic" cell suspension injection model is missing, and/or an explanation for why this was chosen instead of the spheroids used for all other experiments.

We decided to use the intra-striatal injection suspension injection model because of its highly reproducible growth profile and the abundant experience in terms of overall survival. We had used this in the past and it is well established with respect to the dosing of chemotherapy. Therefore it seemed advisable to use this model for our endpoint survival study. We have now added a citation.

This experiment seems somewhat incomplete. Did chemotherapy recruit additional bone-marrow derived cells into the tumor, and were these macrophages or perhaps monocytes/neutrophils? Did aCSF-1 deplete these TMZ-recruited cells? What were the effects of chemotherapy and combination therapy on vasculature? How about tumor volume? Hypoxia? Sprouting dynamics?

We appreciate the interesting questions. However, this experiment is a positive proof of principle intended to investigate whether the aCSF1 induced vascular changes impact on efficacy of chemotherapy. As such, we believe it is conclusive and complete. However, whether or not chemotherapy itself changed the settings by modifying the immune cell populations, and how the two treatments might interact in this aspect would seem to require a separate larger scale study. Nevertheless, we performed additional investigations to gain further insight: In order to check hypoxia in response to the concomitant treatments, we performed Glut1 staining, which indicates that the chemotherapeutic agent treatment together with anti-macrophage treatment significantly reduced tumor hypoxia likely because of an increased tumor oxygenation through vessel normalization. These results are presented in Supplementary Figure 14 C and D.

We further quantified blood vessel diameter in all the conditions and present this in Figure 8D. Combination treatment significantly reduces blood vessel caliber by about 40%. Finally, the dose of TMZ used in the present study did not affect bone marrow cell composition, as indicated by FACS analysis now shown in Supplementary Figure 14E.

It is stated in the manuscript that "cell death" was more homogenous with combination therapy than with chemotherapy alone. This needs to be quantified. An apoptosis marker (activated Caspase 3?) would be a better indicator of cell death than the DNA damage marker pH2AX. Can the cell death observed be correlated to density/"normality" of the vasculature?

As mentioned by the reviewer, activated Caspase-3 staining has been performed and the results, presented in Figure 8 C and E, confirm the phospho-H2AX results.

It is not clear from the methods if multiple comparisons have been corrected for in the p-values for the survival curves, and how?

The quantification indeed results from a multiple comparison test. Quantifications are now more detailed in the method section and in the figure legends.

Inhibition of CSF-1R/macrophage depletion has been shown to improve responses to chemotherapy or radiotherapy in tumor models in several studies (at least DeNardo et al, Cancer Discov 2011; Mitchem et al, Cancer Res 2013; Xu et al, J Urol 2014; Shiao et al, Cancer Immunol Res 2015). Also Hughes et al, Cancer Res 2015 is of interest. This existing literature should be referred to and discussed in this manuscript.

We agree and have now discussed and cited these studies in the manuscript.

3. References and discussion of previous research

The manuscript includes several outdated references, which omit important more recent data. Ref 1 is textbook knowledge and can be left out. Outdated references include Ref4 (replace with e.g. Carmeliet & Jain 2011), Ref 5 (e.g. Noy & Pollard 2014, Qian & Pollard 2010), Ref 7 (replace with e.g. Zumsteg & Christofori 2009, or add another reference such as Coffelt et al 2009, or Qian & Pollard 2010, which also has a full chapter on tumor macrophages and angiogenesis)

Discussion lines 344-346: For completeness, add Cotechini et al Cancer J 2015 review; among other data it has a list of all myeloid-targeted therapeutics currently in clinical trials.

The discussion should include a slightly more extensive treatment of how this manuscript fits into the context of previous data on macrophage depletion, CSF-1/CSF-1R inhibition and angiogenesis. A large body of data proves that macrophages stimulate tumor angiogenesis in a large variety of models, especially in breast cancer models, and most often macrophage depletion and accompanying reduced angiogenesis has led to attenuated tumor growth; myeloid cells are also involved in intra- and extravasation and support invasion and metastasis. In gliomas, myeloid cell depletion (De Palma et al, Cancer Cell 2005; Zhai et al, Glia 2010) or CSF-1R inhibition (Pyonteck et al 2013) also had anti-tumor effects, although acceleration of tumor growth, as in this manuscript, has also been shown before (Galarneau et al, Cancer Res 2011; Stockmann et al 2008).

We thank the reviewer for the efforts in providing these helpful and constructive suggestions. We have now expanded the discussion and added references.

4. Miscellaneous

Results section, lines 231-237: some/all could be moved to the discussion.

Macrophage depletion was only 50% - discuss if a more complete depletion (with anti-CSF-1R?) would have given different results? Were the remaining macrophages M1 or M2 polarized?

Following anti-CSF1 treatment, depleting macrophages, the polarization of the remaining macrophages seems unaffected as shown in Figure 5C.

There is no description in the methods of how tumor volumes were measured.

The complete tumor volume was measured on 200µm serial vibratome sections. This is now mentioned in the methods section.

Glut1 has been used as a pan-endothelial marker in the human glioma samples. Is there no reason to believe that its expression is downregulated in abnormal vessels in high-grade gliomas?

Glut1 was used as a pan-endothelial marker in the human glioma samples because its staining was more reliable than CD31 and endomucin. It is true that its expression decrease in abnormal vessels, but not to an extent that would render it undetectable.

The slight increase in neutrophil recruitment due to CSF-1 inhibition shown in Supplementary Figure 11 should be more directly mentioned in the results and discussed.

After overall quantification revisions, there is only a tendency of neutrophil recruitment following anti-CSF1 mAb treatment. This result is mentioned in the result section and discussed.

Discussion, lines 350-355 are a bit confusing. If IL-34 is not present at all in glioma ("controversial"), CSF-1R inhibition would presumably not have very different effects from CSF-1 inhibition? In any case, inhibition of both ligands (and possible ligand-independent effects?) in a cancer setting is not necessarily a disadvantage.

Inhibition of both ligand might not be a disadvantage indeed if both ligands signal through the same pathway. We wanted here to specify that we cannot exclude that this is not the case and so that we might have different results from CSF1R mAb treatments.

Methods, line 414, title should be Immunofluorescent staining only?

The title has been corrected to "Immunofluorescent staining"

Methods, line 478, use more scientific language.

This sentence has been modified to "Spheroids of 200-250µm were selected for implantation."

The manuscript is well understandable, but grammar checking by a native English speaker would improve it.

The manuscript has now been proof read and edited by a native English speaker, and has hopefully been improved.

Referee 3

Referee #3 (Remarks):

This is a very interesting contribution from a laboratory interested in blood vessel patterning in developing organs. In the present work, the authors investigate tumor angiogenesis, with the goal to establish similarities and differences between normal and tumor vessel patterning. They use sophisticated multiphoton imaging of glioma cells implanted into mice, and show that while the tumor vessels are initially similar to normal vessels with individual tip and stalk cells, over time, they progressively become enlarged and dysfunctional. This abnormalization correlates with recruitment of bone-marrow derived macrophages and a switch from a M1 to a M2 phenotype. Analysis of human glioma samples reveals that both vessel enlargement and M2 macrophage recruitment are remarkably well conserved in human high grade glioblastoma. Blocking macrophage recruitment using anti-CSF1 antibodies prevents blood vessel abnormalization, while recombinant CSF1 treatment of early stage tumors enla

rges tumor vessels. M2 macrophages produce Vegf, and the authors show that genetic blockade of macrophage Vegf production deletion prevents vessel abnormalization. Tumors with better blood vessels grow faster, and are more sensitive to Temozolimid, providing an approach to enhance delivery of cytotoxic agents via improved vasculature. Overall, I find that the work is very well executed, the data are clear and the message is highly interesting to the wide readership of Embo Mol Med. I am in favor of publication but have a number of minor issues the authors should address to improve the presentation of the data.

We thank the reviewer for the very positive appreciation of our work.

Specific comments:

1. The illustration needs a complete overhaul. All the legends of all the graphs are way too small and impossible to read on a printout. Please increase font size.

We agree and the font size has now been increased.

2. Statistic methods should be included in the figure legends.

Statistical methods have been added to the Figure legends.

3. Fig.3. Legend title doesn't make sense. I guess switch is meant rather than in situ.

We are grateful to the reviewer for point this out. This mistake has now been corrected.

4. The blue macrophage staining is hard to see. Could it be changed to a different color to be easier to see?

The blue macrophages were switched to white in order facilitate visualization.

5. The Introduction is very short and does not highlight the novelty of the paper. In my mind, there are three novel aspects that could be mentioned. First, the study directly demonstrates vessel abnormalization by longitudinal imaging. I am not aware of other studies using this methodology to image tumor vessel development, if there are, the authors should cite them! Second, while it is known that macrophages affect tumor progression, this study shows that macrophages affect the vasculature, which again to my knowledge has not been reported before. Third, they show that macrophages induce alterations in Vegf gradients, and that it is the change in the gradient of this factor, rather than the presence of a hypothetical tumor angiogenesis factor that leads to the chaotic nature of the intratumor vasculature.

We thank the referee for the constructive help to improve our manuscript. Modifications have been made in the introduction part to highlight these advances more clearly.

2nd Editorial Decision

29 August 2017

Thank you for the submission of your revised manuscript to EMBO Molecular Medicine. We have now received the enclosed reports from the reviewers that were asked to re-assess it.

As you will see the Reviewers are now satisfied with your manuscript and I am thus prepared to accept your manuscript for publication pending the following editorial amendments:

- 1) We are still missing the author checklist, which I requested in my previous decision letter. Furthermore, please provide 5 keywords, a conflict of interest statement, the running title and 5 keywords
- 2) Please use the appropriate reference list style (<http://embomolmed.embopress.org/authorguide#referencesformat>)
- 3) You have provided 14 EV figures. Please note however that only up to 5, exceptionally 6, supplementary figures can be chosen for inclusion in the article as Expanded View. The remaining

should be included in an Appendix to be provided as a PDF file. The Appendix should begin with a short table of contents. Please refer to our detailed author guidelines (embomolmed.embopress.org/authorguide#expandedview). As a consequence, the manuscript callouts and legends for all supplementary figures (EV and Appendix) will have to be carefully amended where necessary to reflect the correct nomenclature: Appendix figures are referred to in the text as Appendix Figure S1, Appendix Figure S2, etc.

4) Please move the EV legends to the main manuscript file and the Table EV1 header to the table file.

5) EV movie legends must be zipped together with the individual movie file before uploading.

6) Please note that current Fig. EV14 contains panels A-E but the legend describes panels A-D only.

7) As per our Author Guidelines, the description of all reported data that includes statistical testing must state the name of the statistical test used to generate error bars and P values, the number (n) of independent experiments underlying each data point (not replicate measures of one sample), and the actual P value for each test (not merely 'significant' or ' $P < 0.05$ ').

8) We encourage the publication of source data, with the aim of making primary data more accessible and transparent to the reader. Would you be willing to provide a PDF file per figure that contains the original, uncropped and unprocessed scans of all or at least the key gels used in the manuscript and/or source data sets for relevant graphs? The files should be labeled with the appropriate figure/panel number, and in the case of gels, should have molecular weight markers; further annotation may be useful but is not essential. The files will be published online with the article as supplementary "Source Data" files. If you have any questions regarding this just contact me.

For all the above, please contact us in case of difficulties or doubts before re-submission to avoid delaying publication further.

Please submit your revised manuscript within two weeks. I look forward to seeing a revised form of your manuscript as soon as possible.

***** Reviewer's comments *****

Referee #1 (Remarks for Author):

I'm satisfied from the authors' revision

Referee #2 (Remarks for Author):

The authors have satisfactorily answered my concerns and they have made very professional revision of their submission, which should now be accepted for publication.

2nd Revision - authors' response

12 September 2017

Authors made requested editorial changes.

Corresponding Author Name: Holger Gerhardt
Journal Submitted to: EMBO Molecular Medicine
Manuscript Number: EMM-2016-07445-V2